# Rethinking Channel Dependence for Multivariate Time Series Forecasting: Learning from Leading Indicators

**Lifan Zhao**
Shanghai Jiao Tong University
`mogician233@sjtu.edu.cn`

**Yanyan Shen**[*]
Shanghai Jiao Tong University
`shenyy@sjtu.edu.cn`

## Abstract

Recently, channel-independent methods have achieved state-of-the-art performance in multivariate time series (MTS) forecasting. Despite reducing overfitting risks, these methods miss potential opportunities in utilizing channel dependence for accurate predictions. We argue that there exist locally stationary lead-lag relationships between variates, i.e., some lagged variates may follow the leading indicators within a short time period. Exploiting such channel dependence is beneficial since leading indicators offer advance information that can be used to reduce the forecasting difficulty of the lagged variates. In this paper, we propose a new method named LIFT that first efficiently estimates leading indicators and their leading steps at each time step and then judiciously allows the lagged variates to utilize the advance information from leading indicators. LIFT plays as a plugin that can be seamlessly collaborated with arbitrary time series forecasting methods. Extensive experiments on six real-world datasets demonstrate that LIFT improves the state-of-the-art methods by 5.4% in average forecasting performance. Our code is available at `https://github.com/SJTU-Quant/LIFT`.

## 1 Introduction

Multivariate time series (MTS) forecasting, one of the most popular research topics, is a fundamental task in various domains such as weather, traffic, and finance. An MTS consists of multiple channels (*a.k.a.*, variates[1]), where each channel is a univariate time series. Many MTS forecasting researches argue each channel has dependence on other channels. Accordingly, numerous approaches adopt *channel-dependent* (CD) strategies and *jointly* model multiple variates by advanced neural architectures, including GNNs (Wu et al., 2020; Cao et al., 2020; Huang et al., 2023; Yi et al., 2023a), MLPs (Chen et al., 2023; Ekambaram et al., 2023; Wang et al., 2024a; Yi et al., 2023b), CNNs (Wu et al., 2023), Transformers (Zhou et al., 2021; Ni et al., 2023; Wang et al., 2024b; Liu et al., 2023a), and others (Shen et al., 2024; Jia et al., 2023; Fan et al., 2024).

Unexpectedly, CD methods have been defeated by recently proposed channel-independent (CI) methods (Nie et al., 2023; Lee et al., 2024; Zhou et al., 2023; Jin et al., 2024; Cao et al., 2024; Chen et al., 2024; Dai et al., 2024) and even a simple linear model (Zeng et al., 2023; Li et al., 2023a; Xu et al., 2023). These CI methods *seperately* forecast each univariate time series based on its own historical values, instead of referring to other variates. While only modeling cross-time dependence, CI Transformers (Nie et al., 2023; Zhou et al., 2023) surprisingly outperform CD Transformers that jointly model cross-time and cross-variate dependence (Grigsby et al., 2021; Zhang & Yan, 2023). One reason is that existing CD methods lack prior knowledge about channel dependence and may encounter the overfitting issue (Han et al., 2023). This gives rise to an interesting question: *is there any explicit channel dependence that is effective to MTS forecasting*?

In this work, we turn the spotlight on the **locally stationary lead-lag relationship** between variates. An intriguing yet underestimated characteristic of many MTS is that the evolution of variates may

---

[*]corresponding author.

[1]We use the terms "*variate*" and "*channel*" interchangeably.

Figure 1: Illustration of locally stationary lead-lag relationships. (a) On training data, three variates $v_1$, $v_2$, and $v_3$ share similar temporal patterns (see colors) across the lookback window and horizon window, while $v_1$ and $v_2$ run ahead of $v_3$ by four and two steps, respectively. However, the leading indicators and leading steps can only keep static for a short period. (b) On test data, $v_1$ is no longer a leading indicator, and $v_2$ also changes its leading steps to five.

lag behind some other variates, termed as leading indicators. Leading indicators may directly influence the wave of other variates, while the influence requires a certain time delay to propagate and take effect. For example, an increasing concentration of an anti-fever drug in the blood may cause a decrease in body temperature after an hour but not immediately. On top of this, another common case is that both leading indicators and lagged variates depend on some latent factors, while the leading ones are the first to get affected. For example, a typhoon first cools down coastal cities and, after a few days, cools down inland cities.

As such effects typically change little within a certain period, the lead-lag relationships are locally stationary once established. As illustrated in Figure 1a, the lagged variate and its leading indicators share similar temporal patterns across the lookback window and the horizon window. If a leading indicator evolves $\delta$-step ahead of the target variate, the latest $\delta$ steps of its lookback window will share similar temporal patterns with the future $\delta$ steps of the lagged variate. Particularly, when the lagged variate completely follows its leading indicator, the difficulty of forecasting $H$ steps for the lagged variate can be reduced to forecasting $H - \delta$ steps by previewing the advance information. Despite the advent of lead-lag relationships, **the dynamic variation in leading indicators and leading steps** poses the challenge to modeling channel dependence. As shown in Figure 1, the specific leading indicators and the corresponding leading steps can vary over time.

In light of this, we propose a method named LIFT (short for *Learning from **L**eading **I**ndicators **F**or MTS **F**orecasting*), involving three key steps. *First*, we develop an efficient cross-correlation computation algorithm to dynamically estimate the leading indicators and the leading steps at each time step. *Second*, as depicted in Figure 2, we align each variate and its leading indicators via a target-oriented shift trick. *Third*, we employ a backbone to make preliminary predictions and introduce a Lead-aware Refiner to calibrate the rough predictions. It is noteworthy that many MTS are heterogeneous, where the variates are different dimensions of an object (*e.g.*, wind speed, humidity, and air temperature in weather). In these cases, the lagged variates may be correlated with the leading indicators by sharing only a part of temporal patterns. To address this issue, we exploit desirable signals in the frequency domain and realize the Lead-aware Refiner by an Adaptive Frequency Mixer that adaptively filters out undesirable frequency components of leading indicators and absorbs the remaining desirable ones. The main contributions of this paper are summarized as follows.

- We propose a novel method called LIFT that exploits the locally stationary lead-lag relationship between variates for MTS forecasting. LIFT works as a plug-and-play module and can seamlessly incorporate arbitrary time series forecasting backbones.

- We introduce an efficient algorithm to estimate the leading indicators and the corresponding leading steps at any time step. We further devise a Lead-aware Refiner that adaptively leverages the informative signals of leading indicators in the frequency domain to refine the predictions of lagged variates.

- Extensive experimental results on six real-world datasets demonstrate that LIFT significantly improves the state-of-the-art methods in both short-term and long-term MTS forecasting. Specifically, LIFT makes an average improvement of **7.9%** over CI models and **3.0%** over CD models. We also introduce a lightweight yet strong method LightMTS, which enjoys high parameter efficiency and achieves the best performance on popular Weather and Electricity datasets.

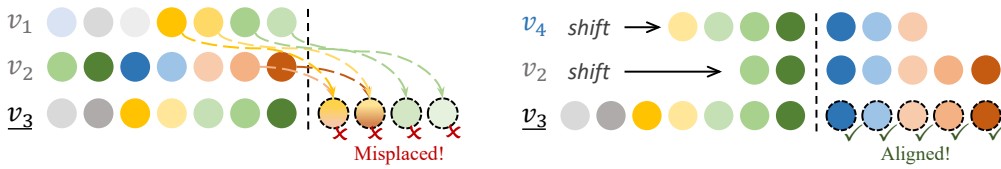

(a) Overfitting training patterns in Figure 1a     (b) Proposed target-oriented shift

Figure 2: Illustration of our key idea. In one case of test data, $v_1$ no longer leads $v_3$. Instead, the leading indicators of $v_3$ are $v_2$ and $v_4$, which lead by five and three steps, respectively. An intuitive idea is to shift $v_2$ and $v_4$ by the corresponding leading steps to keep them always aligned with $v_3$.

## 2 PRELIMINARIES

A multivariate time series (MTS)[2] is denoted by $\boldsymbol{\mathcal{X}} = \left\{\mathcal{X}^{(1)}, \cdots, \mathcal{X}^{(C)}\right\}$, where $C$ is the number of variates (*a.k.a.* channels) and $\mathcal{X}^{(j)}$ is the time series of the $j$-th variate. Given an $L$-length lookback window $\boldsymbol{\mathcal{X}}_{t-L+1:t} = \{\mathcal{X}^{(j)}_{t-L+1}, \cdots, \mathcal{X}^{(j)}_t\}^C_{j=1} \in \mathbb{R}^{C \times L}$, the MTS forecasting task at time $t$ aims to predict $H$ consecutive future time steps in the horizon window, *i.e.*, $\boldsymbol{\mathcal{X}}_{t+1:t+H} \in \mathbb{R}^{C \times H}$.

We assume $\mathcal{X}^{(j)}_{t+1:t+H}$ is similar to $\mathcal{X}^{(i)}_{t+1-\delta:t+H-\delta}$ if variate $i$ leads variate $j$ by $\delta$ steps at time $t$. Through the lens of locally stationary lead-lag relationships, one can use recent observations to estimate the leading indicators and the leading steps. Specifically, the lead-lag relationship can be quantified by the *cross-correlation coefficient* between $\mathcal{X}^{(i)}_{t-L+1-\delta:t-\delta}$ and $\mathcal{X}^{(j)}_{t-L+1:t}$, which is defined as follows.

**Definition 1** (Cross-correlation coefficient). *Assuming variate $i$ is $\delta$ steps ahead of variate $j$ over the $L$-length lookback window, the cross-correlation coefficient between the two variates at time $t$ is defined as:*

$$R^{(j)}_{i,t}(\delta) = \frac{\mathrm{Cov}(\mathcal{X}^{(i)}_{t-L+1-\delta:t-\delta}, \mathcal{X}^{(j)}_{t-L+1:t})}{\sigma^{(i)}\sigma^{(j)}} = \frac{1}{L}\sum_{t'=t-L+1}^{t} \frac{\mathcal{X}^{(i)}_{t'-\delta} - \mu^{(i)}}{\sigma^{(i)}} \cdot \frac{\mathcal{X}^{(j)}_{t'} - \mu^{(j)}}{\sigma^{(j)}}, \quad (1)$$

*where $\mu^{(\cdot)} \in \mathbb{R}$ and $\sigma^{(\cdot)} \in \mathbb{R}$ represent the mean and standard variation of the univariate time series within the lookback window, respectively.*

## 3 THE LIFT APPROACH

In this section, we propose our LIFT method that dynamically identifies leading indicators and adaptively leverages them for MTS forecasting.

### 3.1 OVERVIEW

Figure 3 depicts the overview of LIFT, which involves 6 major steps as follows.

(1) **Preliminary forecasting.** Given a lookback window $\boldsymbol{\mathcal{X}}_{t-L+1:t}$, we first obtain rough predictions $\widehat{\boldsymbol{\mathcal{X}}}_{t+1:t+H}$ from a black-box backbone, which can be implemented by any existing time series forecasting model.

(2) **Instance normalization.** Given $\boldsymbol{\mathcal{X}}_{t-L+1:t}$ and $\widehat{\boldsymbol{\mathcal{X}}}_{t+1:t+H}$, we apply instance normalization (Kim et al., 2022) without affine parameters so as to unify the value range across the variates. Specifically, based on the mean and standard deviation of each variate in $\boldsymbol{\mathcal{X}}_{t-L+1:t}$, we obtain a normalized lookback window $\boldsymbol{X}_{t-L+1:t}$ and normalized predictions $\widehat{\boldsymbol{X}}_{t+1:t+H}$.

(3) **Lead estimation.** Given $\boldsymbol{X}_{t-L+1:t}$, the Lead Estimator calculates the cross-correlation coefficients for pair-wise variates. For each variate $j$, we select the $K$ most possible leading indicators $\mathcal{I}^{(j)}_t \in \mathbb{R}^K$ ($K \ll C$) along with the corresponding leading steps $\{\delta^{(j)}_{i,t} \mid i \in \mathcal{I}^{(j)}_t\}$ and cross-correlation coefficients $\boldsymbol{R}^{(j)}_t \in \mathbb{R}^K$.

---
[2]We use bold symbols to denote matrices of multiple variates.

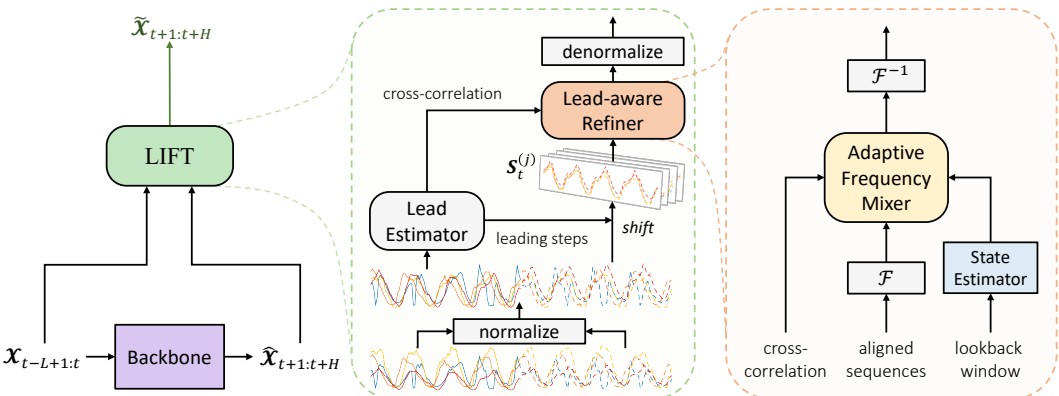

Figure 3: Overview of LIFT. All layers in the grey background are non-parametric. We depict the input of the lookback window by solid curves and the predictions of the horizon window by dashed curves. As an illustration, we choose the two most possible leading indicators for each target variate, *e.g.*, the orange and the yellow ones are leading indicators of the red at time $t$.

(4) **Target-oriented shifts.** After obtaining $\mathcal{I}_t^{(j)}$ and $\{\delta_{i,t}^{(j)}\}_{i \in \mathcal{I}_t^{(j)}}$ for variate $j$, we shift $\boldsymbol{X}_{t-L+1:t}^{(i)}$ and $\widehat{\boldsymbol{X}}_{t+1:t+H}^{(i)}$ by $\delta_{i,t}^{(j)}$ steps where $i \in \mathcal{I}_t^{(j)}$. We thereby obtain a $j$-oriented MTS segment $\boldsymbol{S}_t^{(j)} \in \mathbb{R}^{K \times H}$, where the $K$ leading indicators get aligned with variate $j$ in the horizon window.

(5) **Lead-aware refinement.** The Lead-aware Refiner extracts signals from $\boldsymbol{S}_t^{(j)}$ and refines the normalized preliminary predictions $\widehat{\boldsymbol{X}}_{t+1:t+H}^{(j)}$ as $\widetilde{\boldsymbol{X}}_{t+1:t+H}^{(j)}$.

(6) **Instance denormalization.** Finally, we denormalize $\widetilde{\boldsymbol{X}}_{t+1:t+H}^{(j)}$ with the original mean and standard deviation, yielding the final predictions $\widetilde{\boldsymbol{\mathcal{X}}}_{t+1:t+H}^{(j)}$.

**Training scheme.** We can jointly train the backbone and Lead-aware Refiner by the MSE between $\widetilde{\boldsymbol{\mathcal{X}}}_{t+1:t+H}$ and the ground truth $\boldsymbol{\mathcal{X}}_{t+1:t+H}$. Alternatively, given a pretrained and frozen backbone, we can precompute the preliminary predictions only once on training data, reducing the time of hyperparameter tuning and GPU memory occupation during training.

**Technical challenges.** Notably, it is non-trivial to leverage the lead-lag relationships due to issues of efficiency and noise. As Eq. (1) requires $\mathcal{O}(L)$, a brute-force estimation method that searches all possible $\delta$ in $\{1, \cdots, L\}$ requires $\mathcal{O}(L^2)$ computations. Also, $\boldsymbol{S}_t^{(j)}$ can contain some irrelevant patterns from leaders which are noise to the lagged variate.

To tackle these issues, we implement the Lead Estimator by an efficient algorithm of $\mathcal{O}(L \log L)$ complexity. And we develop the Lead-aware Refiner by an Adaptive Frequency Mixer that adaptively generates frequency-domain filters and mixes desirable frequency components according to the cross-correlations and variate states.

## 3.2 LEAD ESTIMATOR

Given the normalized lookback window $\boldsymbol{X}_{t-L+1:t}$, the Lead Estimator first computes the cross-correlation coefficients between each pair of variate $i$ and variate $j$, based on an extension of Wiener–Khinchin theorem (Wiener, 1930) (see details in Appendix A). Formally, we estimate the coefficients for all possible leading steps in $\{0, \cdots, L-1\}$ at once by the following equation:

$$\left\{R_{i,t}^{(j)}(\tau)\right\}_{\tau=0}^{L-1} = \frac{1}{L}\mathcal{F}^{-1}\left(\mathcal{F}(X_{t-L+1:t}^{(j)}) \odot \overline{\mathcal{F}(X_{t-L+1:t}^{(i)})}\right), \qquad (2)$$

where $\mathcal{F}$ is the Fast Fourier Transform, $\mathcal{F}^{-1}$ is its inverse, $\odot$ is the element-wise product, and the bar denotes the conjugate operation. The complexity is reduced to $\mathcal{O}(L \log L)$.

Note that variates can exhibit either positive or negative correlations. The leading step $\delta_{i,t}^{(j)}$ between the target variate $j$ and its leading indicator $i$ is meant to reach the maximum absolute cross-

correlation coefficient, *i.e.*,

$$\delta_{i,t}^{(j)} = \arg\max_{1 \leq \tau \leq L-1} |R_{i,t}^{(j)}(\tau)|. \tag{3}$$

For simplicity, we denote the maximum absolute coefficient $|R_{i,t}^{(j)}(\delta_{i,t}^{(j)})|$ as $|R_{i,t}^{(j)*}|$. Then, we choose $K$ variates that show the most significant lead-lag relationships as leading indicators of variate $j$, which are defined as:

$$\mathcal{I}_t^{(j)} = \arg\operatorname*{TopK}_{1 \leq i \leq C}(|R_{i,t}^{(j)*}|). \tag{4}$$

Specifically, the $K$ leading indicators $\mathcal{I}_t^{(j)}$ are sorted by cross-correlations in descending order, *i.e.*, the $k$-th indicator in $\mathcal{I}_t^{(j)}$ has the $k$-th highest $|R_{i,t}^{(j)*}|$ *w.r.t.* variate $j$. Furthermore, we use $\boldsymbol{R}_t^{(j)} \in \mathbb{R}^K$ to denote an array of $\{|R_{i,t}^{(j)*}|\}_{i \in \mathcal{I}_t^j}$.

Notably, our Lead Estimator is non-parametric and we can precompute the estimations only once on training data, instead of repeating the computations at every epoch.

### 3.3 LEAD-AWARE REFINER

For each variate $j$, the Lead-aware Refiner is to refine $\widehat{\mathcal{X}}_{t+1:t+H}^{(j)}$ by its leading indicators. We will describe the refinement process for variate $j$, and the other $C-1$ variates are refined in parallel.

**Target-oriented shifts**  For each leading indicator $i \in \mathcal{I}_t^{(j)}$, we shift its sequence by the leading step as follows:

$$\boldsymbol{X}_{t+1:t+H}^{(i \to j)} = \begin{cases} \boldsymbol{X}_{t+1-\delta_{i,t}^{(j)}:\, t+H-\delta_{i,t}^{(j)}}^{(i)}, & \text{if } \delta_{i,t}^{(j)} \geq H \\ \boldsymbol{X}_{t+1-\delta_{i,t}^{(j)}:\, t}^{(i)} \| \widehat{\boldsymbol{X}}_{t+1:\, t+H-\delta_{i,t}^{(j)}}^{(i)}, & \text{otherwise} \end{cases} \tag{5}$$

where $\|$ is the concatenation.

For a leading indicator $i$ that is negatively correlated with the variate $j$, we flip its values at each time step to reflect $R_{i,t}^{(j)*} < 0$. Formally, for each $i \in \mathcal{I}_i^{(j)}$, we have:

$$\operatorname{turn}(\boldsymbol{X}_{t+1:t+H}^{(i \to j)}) = \operatorname{sign}(R_{i,t}^{(j)*}) \cdot \boldsymbol{X}_{t+1:t+H}^{(i \to j)}. \tag{6}$$

We then collect $\{\operatorname{turn}(\boldsymbol{X}_{t+1:t+H}^{(i \to j)}) \mid i \in \mathcal{I}_t^{(j)}\}$ as a target-oriented MTS segment $\boldsymbol{S}_t^{(j)} \in \mathbb{R}^{K \times H}$.

**State estimation**  For a comprehensive understanding of leading indicators, it is noteworthy that the lead-lag patterns also depend on variate states. Different variates lie in their specific states with some intrinsic periodicities (or trends), *e.g.*, solar illumination is affected by rains in the short term but keeps its daily periodicity. The state of a variate may also change over time, exhibiting different correlation strengths with other variates, *e.g.*, correlations between the traffic speeds of two adjacent roads are strong within peak hours but much weaker within off-peak hours. Therefore, the variate states are informative signals that can guide us to filter out uncorrelated patterns.

Assuming there are $N$ states in total, we estimate the state probabilities of variate $j$ at time $t$ by:

$$P_t^{(j)} = \operatorname{softmax}\left(P_0^{(j)} + f_{\text{state}}(\mathcal{X}_{t-L+1:t}^{(j)})\right), \tag{7}$$

where $P_0^{(j)} \in \mathbb{R}^N$ represents the intrinsic state distribution of variate $j$ and is a learnable parameter, $f_{\text{state}} : \mathbb{R}^L \mapsto \mathbb{R}^N$ is implemented by a linear layer, and $P_t^{(j)} = \{p_{t,n}^{(j)}\}_{n=1}^N \in \mathbb{R}^N$ includes the probabilities of all potential states at time $t$. Our adaptive frequency mixer will take $P_t^{(j)}$ to generate filters to filter out noisy channel dependence according to the variate state.

**Adaptive frequency mixer**  To extract valuable information from leading indicators, we propose to model cross-variate dependence in the frequency domain. Given the normalized predictions of variate $j$ and its target-oriented MTS segment $\boldsymbol{S}_t^{(j)}$, we derive their Fourier transforms by:

$$V^{(j)} = \mathcal{F}(\widehat{\boldsymbol{X}}_{t+1:t+H}^{(j)}) \quad \text{and} \quad \boldsymbol{U}^{(j)} = \mathcal{F}(\boldsymbol{S}_t^{(j)}), \tag{8}$$

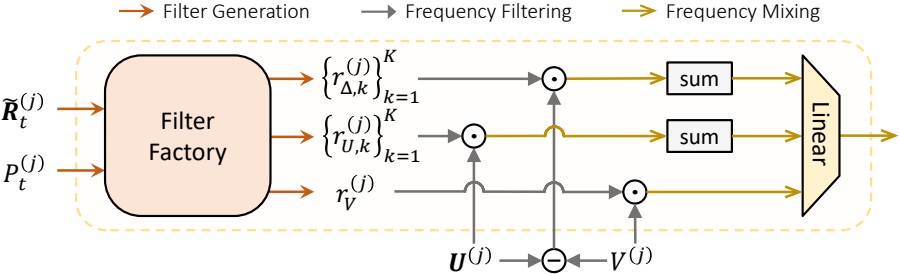

Figure 4: Architecture of the adaptive frequency mixer.

where $\mathcal{F}$ is the Fast Fourier Transform, $V^{(j)} \in \mathbb{C}^{\lfloor H/2 \rfloor + 1}$, and $\boldsymbol{U}^{(j)} \in \mathbb{C}^{K \times (\lfloor H/2 \rfloor + 1)}$. Each element of $\boldsymbol{U}^{(j)}$, denoted as $U_k^{(j)}$, is the frequency components of the $k$-th leading indicator. Let $\Delta_k^{(j)} = U_k^{(j)} - V^{(j)}$ denote the difference between variate $j$ and the $k$-th leading indicator.

Intuitively, the preliminary predictions deserve more refinement from the leading indicators when the estimated correlation $\boldsymbol{R}_t^{(j)}$ is large. To filter signals in $V^{(j)}$ and $\boldsymbol{U}^{(j)}$, we employ a filter factory to generate $2K + 1$ frequency-domain filters as defined below:

$$[r_{U,1}^{(j)}, \cdots, r_{U,K}^{(j)}, r_{\Delta,1}^{(j)}, \cdots, r_{\Delta,K}^{(j)}, r_V^{(j)}] = \sum\nolimits_{n=1}^{N} p_n^{(j)} \cdot f_n(\boldsymbol{R}_t^{(j)}), \tag{9}$$

where $f_n : \mathbb{R}^K \mapsto \mathbb{R}^{(2K+1)(\lfloor H/2 \rfloor + 1)}$ is a linear layer with parameters specific to the $n$-th state. On the one hand, we use the first $2K$ filters to model two kinds of lead-lag relationships: (1) variate $j$ is directly influenced by the $k$-th leader, and the ground-truth $V_{true}^{(j)}$ contains a degree of $U_k^{(j)}$, e.g., $V_{true}^{(j)} \approx V^{(j)} + r_{U,k}^{(j)} \odot U_k^{(j)}$; (2) variate $j$ is similar to the $k$-th leader when they are both influenced by a latent factor, and the ground-truth $V_{true}^{(j)}$ is the interpolation between $V^{(j)}$ and $U_k^{(j)}$, e.g., $V_{true}^{(j)} \approx (1 - r_{\Delta,k}^{(j)}) \odot V^{(j)} + r_{\Delta,k}^{(j)} \odot U = V^{(j)} + r_{\Delta,k}^{(j)} \odot \Delta_k^{(j)}$. On the other hand, we use $r_V^{(j)} \in \mathbb{R}^{\lfloor H/2 \rfloor + 1}$ to dismiss unreliable frequency components of $V^{(j)}$. Formally, we scale the frequency components by:

$$\widetilde{V}^{(j)} = r_V^{(j)} \odot V^{(j)}, \quad \widetilde{U}_k^{(j)} = r_{U,k}^{(j)} \odot U_k^{(j)}, \quad \widetilde{\Delta}_k^{(j)} = r_{\Delta,k}^{(j)} \odot \Delta_k^{(j)}. \tag{10}$$

Then, we gather information from $K$ leading indicators and mix the frequency components by:

$$\widetilde{V}^{(j)} = g\left(\widetilde{V}^{(j)} \parallel \sum\nolimits_{k=1}^{K} \widetilde{U}_k^{(j)} \parallel \sum\nolimits_{k=1}^{K} \widetilde{\Delta}_k^{(j)}\right), \tag{11}$$

where $g : \mathbb{C}^{3(\lfloor H/2 \rfloor + 1)} \mapsto \mathbb{C}^{\lfloor H/2 \rfloor + 1}$ is a complex-valued linear layer.

Finally, we apply inverse Fast Fourier Transform and denormalization in order to derive the final refined predictions, which are formulated as:

$$\tilde{\mathcal{X}}_{t+1:t+H}^{(j)} = \text{denorm}(\mathcal{F}^{-1}(\widetilde{V}^{(j)})), \tag{12}$$

where we use the mean and standard deviation of $\mathcal{X}_{t-L+1:t}^{(j)}$ for denormalization.

### 3.4 DISCUSSION

**Reasoning why CD models show inferior performance.** Many variates are unaligned with each other, while traditional models (e.g., Informer (Zhou et al., 2021)) simply mix multivariate information at the same time step. Consequently, they introduce outdated information from lagged variates which are noise and disturb predicting leaders. Though other models (e.g., Vector Auto-Regression (Giannone et al., 2010)) memorize CD from different time steps by static weights, they can suffer from overfitting issues since the leading indicators and leading steps vary over time.

**LIFT can cooperate with arbitrary time series forecasting backbones.** When combining LIFT with a CI backbone, we decompose MTS forecasting into two stages which focus on modeling time

dependence and channel dependence, respectively. This scheme avoids introducing noisy channel dependence during the first stage and may reduce optimization difficulty compared with traditional CD methods. When combining LIFT with a CD backbone, we expect LIFT to refine the rough predictions with the actual observations of leading indicators in $\boldsymbol{S}_t^{(j)}$.

**LIFT alleviates distribution shifts by dynamically selecting and shifting indicators.** Existing normalization-based methods (Kim et al., 2022; Fan et al., 2023; Liu et al., 2023b) handle distribution shifts of the statistical properties (*e.g.*, mean and variance) in the lookback window and the horizon window. Our work is orthogonal to them as we take a novel investigation into a different kind of distribution shifts in channel dependence (see visualization in Appendix D.2).

## 4 LIGHTWEIGHT MTS FORECASTING WITH LIFT

Thanks to the flexibility of LIFT, we introduce a lightweight MTS forecasting method named LightMTS, where a simple linear layer serves as a CI backbone. Following Li et al. (2023a), we conduct instance normalization before preliminary forecasting to alleviate distribution shifts.

As we do not learn representations in the high-dimensional latent space, LightMTS is more lightweight than popular CD models, including Transformers (Zhang & Yan, 2023; Liu et al., 2023a) and CNNs (Wu et al., 2023). Empirical evidence is provided in Appendix D.1, where the parameter efficiency of LightMTS keeps similar to DLinear Zeng et al. (2023).

## 5 EXPERIMENTS

### 5.1 EXPERIMENTAL SETTINGS

**Datasets.** We conduct extensive experiments on six widely-used MTS datasets, including Weather (Zeng et al., 2023), Electricity (Wu et al., 2020), Traffic (Lai et al., 2018), Solar (Liu et al., 2023a), Wind (Liu et al., 2022), and PeMSD8 (Song et al., 2020). We provide the dataset details in Appendix C.1 and conduct experiments on more datasets in Appendix D.3.

**Comparison Methods.** As LIFT can incorporate arbitrary time series forecasting backbones, we verify the effectiveness of LIFT with (i) *two state-of-the-art CI models*: PatchTST (Nie et al., 2023) and DLinear (Zeng et al., 2023); (ii) *the state-of-the-art CD model*: Crossformer (Zhang & Yan, 2023); (iii) *a classic CD model*: MTGNN (Wu et al., 2020). We use them to instantiate the backbone of LIFT, while we keep the same model hyperparameters for fair comparison. We also include the baselines of PatchTST, such as FEDformer (Zhou et al., 2022) and Autoformer (Wu et al., 2021).

**Setups.** All of the methods follow the same experimental setup with the forecast horizon $H \in \{24, 48, 96, 192, 336, 720\}$ for both short-term and long-term forecasting. We collect some baseline results reported by PatchTST to compare performance with LightMTS, where PatchTST has tuned the lookback length $L$ of FEDformer and Autoformer. For other methods, we set $L$ to 336. We use Mean Squared Error (MSE) and Mean Absolute Error (MAE) as evaluation metrics.

### 5.2 PERFORMANCE EVALUATION

Table 1 compares the forecasting performance between the four state-of-the-art methods and LIFT on the six MTS datasets, showing that LIFT can outperform the SOTA methods in most cases. Specifically, LIFT improves the corresponding backbone by 5.4% on average.

**Improvement over CI Backbones.** LIFT makes an average improvement of 7.9% over PatchTST and DLinear on the six datasets. Notably, PatchTST and DLinear surpass Crossformer and MTGNN by a large margin on Weather, Electricity, and Traffic datasets, indicating the challenge of modeling channel dependence. Intriguingly, LIFT significantly improves CI backbones by an average margin of 4.7% on these challenging datasets, achieving the best performance in most cases. This confirms that LIFT can reduce overfitting risks by introducing prior knowledge about channel dependence.

**Improvement over CD Backbones.** LIFT makes an average improvement of 3.0% over Crossformer and MTGNN on the six datasets. As CD backbones outperform CI ones on Solar, Wind, and

Table 1: Performance comparison in terms of forecasting errors. We highlight the better results between each pair of backbones and LIFT in **bold** and the best results among all methods on each dataset with underlines. We show the relative improvement of LIFT over the corresponding backbone in the rightmost column.

| Method | | PatchTST MSE | MAE | + LIFT MSE | MAE | DLinear MSE | MAE | + LIFT MSE | MAE | Crossformer MSE | MAE | + LIFT MSE | MAE | MTGNN MSE | MAE | + LIFT MSE | MAE | Impr. |
|---|---|---|---|---|---|---|---|---|---|---|---|---|---|---|---|---|---|---|
| Weather | 24 | 0.091 | 0.122 | 0.089 | 0.119 | 0.104 | 0.152 | 0.090 | 0.125 | 0.086 | 0.126 | 0.086 | 0.126 | 0.090 | 0.128 | 0.090 | 0.126 | 4.7% |
| | 48 | 0.119 | 0.164 | 0.115 | 0.158 | 0.137 | 0.194 | 0.114 | 0.163 | 0.112 | 0.166 | 0.112 | 0.165 | 0.117 | 0.170 | 0.115 | 0.167 | 5.3% |
| | 96 | 0.152 | 0.199 | 0.146 | 0.196 | 0.176 | 0.237 | 0.145 | 0.203 | 0.145 | 0.209 | 0.146 | 0.210 | 0.157 | 0.216 | 0.154 | 0.212 | 5.0% |
| | 192 | 0.197 | 0.243 | 0.190 | 0.238 | 0.220 | 0.282 | 0.189 | 0.249 | 0.197 | 0.264 | 0.196 | 0.262 | 0.205 | 0.269 | 0.203 | 0.266 | 4.3% |
| | 336 | 0.249 | 0.283 | 0.243 | 0.281 | 0.265 | 0.319 | 0.243 | 0.292 | 0.246 | 0.309 | 0.245 | 0.305 | 0.258 | 0.312 | 0.256 | 0.308 | 3.0% |
| | 720 | 0.320 | 0.335 | 0.315 | 0.333 | 0.323 | 0.362 | 0.317 | 0.349 | 0.323 | 0.364 | 0.321 | 0.360 | 0.335 | 0.369 | 0.333 | 0.365 | 1.4% |
| Electricity | 24 | 0.099 | 0.196 | 0.094 | 0.190 | 0.110 | 0.209 | 0.099 | 0.197 | 0.095 | 0.195 | 0.093 | 0.193 | 0.097 | 0.195 | 0.094 | 0.193 | 3.6% |
| | 48 | 0.115 | 0.210 | 0.110 | 0.205 | 0.125 | 0.223 | 0.113 | 0.209 | 0.116 | 0.216 | 0.113 | 0.211 | 0.116 | 0.215 | 0.112 | 0.211 | 4.0% |
| | 96 | 0.130 | 0.222 | 0.128 | 0.222 | 0.140 | 0.237 | 0.130 | 0.225 | 0.142 | 0.243 | 0.138 | 0.238 | 0.138 | 0.238 | 0.133 | 0.233 | 2.9% |
| | 192 | 0.148 | 0.240 | 0.147 | 0.239 | 0.153 | 0.249 | 0.148 | 0.242 | 0.159 | 0.259 | 0.154 | 0.251 | 0.160 | 0.261 | 0.153 | 0.252 | 2.7% |
| | 336 | 0.167 | 0.261 | 0.163 | 0.257 | 0.169 | 0.267 | 0.163 | 0.261 | 0.192 | 0.293 | 0.176 | 0.276 | 0.193 | 0.284 | 0.187 | 0.275 | 3.8% |
| | 720 | 0.202 | 0.291 | 0.195 | 0.289 | 0.203 | 0.301 | 0.198 | 0.295 | 0.264 | 0.353 | 0.224 | 0.312 | 0.242 | 0.327 | 0.216 | 0.305 | 6.6% |
| Traffic | 24 | 0.323 | 0.235 | 0.300 | 0.214 | 0.371 | 0.267 | 0.347 | 0.255 | 0.483 | 0.273 | 0.392 | 0.246 | 0.402 | 0.260 | 0.392 | 0.259 | 7.3% |
| | 48 | 0.342 | 0.240 | 0.329 | 0.236 | 0.393 | 0.276 | 0.367 | 0.260 | 0.513 | 0.290 | 0.428 | 0.289 | 0.450 | 0.274 | 0.436 | 0.281 | 4.4% |
| | 96 | 0.367 | 0.251 | 0.352 | 0.242 | 0.410 | 0.282 | 0.394 | 0.273 | 0.519 | 0.293 | 0.462 | 0.284 | 0.479 | 0.289 | 0.464 | 0.286 | 4.2% |
| | 192 | 0.385 | 0.259 | 0.373 | 0.251 | 0.423 | 0.287 | 0.413 | 0.281 | 0.522 | 0.296 | 0.490 | 0.283 | 0.507 | 0.307 | 0.491 | 0.301 | 3.3% |
| | 336 | 0.398 | 0.265 | 0.389 | 0.262 | 0.436 | 0.296 | 0.426 | 0.288 | 0.530 | 0.300 | 0.517 | 0.303 | 0.539 | 0.314 | 0.519 | 0.309 | 1.9% |
| | 720 | 0.434 | 0.287 | 0.429 | 0.286 | 0.466 | 0.315 | 0.454 | 0.307 | 0.584 | 0.369 | 0.543 | 0.322 | 0.616 | 0.352 | 0.532 | 0.340 | 5.4% |
| Solar | 24 | 0.095 | 0.160 | 0.087 | 0.147 | 0.133 | 0.219 | 0.093 | 0.149 | 0.082 | 0.134 | 0.079 | 0.129 | 0.070 | 0.125 | 0.069 | 0.122 | 11.0% |
| | 48 | 0.153 | 0.227 | 0.143 | 0.200 | 0.190 | 0.267 | 0.145 | 0.197 | 0.146 | 0.203 | 0.140 | 0.178 | 0.131 | 0.180 | 0.130 | 0.177 | 11.0% |
| | 96 | 0.176 | 0.227 | 0.174 | 0.224 | 0.222 | 0.291 | 0.185 | 0.238 | 0.179 | 0.245 | 0.174 | 0.224 | 0.167 | 0.224 | 0.166 | 0.223 | 6.2% |
| | 192 | 0.205 | 0.260 | 0.190 | 0.245 | 0.249 | 0.309 | 0.194 | 0.253 | 0.204 | 0.254 | 0.197 | 0.250 | 0.180 | 0.243 | 0.179 | 0.239 | 7.6% |
| | 336 | 0.200 | 0.252 | 0.194 | 0.249 | 0.269 | 0.324 | 0.198 | 0.260 | 0.216 | 0.257 | 0.204 | 0.254 | 0.191 | 0.251 | 0.190 | 0.245 | 7.5% |
| | 720 | 0.229 | 0.282 | 0.203 | 0.261 | 0.271 | 0.327 | 0.207 | 0.260 | 0.211 | 0.250 | 0.202 | 0.255 | 0.197 | 0.256 | 0.195 | 0.251 | 8.5% |
| Wind | 24 | 0.137 | 0.179 | 0.131 | 0.175 | 0.151 | 0.198 | 0.136 | 0.182 | 0.122 | 0.173 | 0.121 | 0.168 | 0.124 | 0.172 | 0.124 | 0.170 | 3.8% |
| | 48 | 0.163 | 0.200 | 0.155 | 0.196 | 0.175 | 0.214 | 0.159 | 0.200 | 0.147 | 0.194 | 0.147 | 0.189 | 0.149 | 0.192 | 0.148 | 0.191 | 3.4% |
| | 96 | 0.186 | 0.216 | 0.175 | 0.213 | 0.197 | 0.230 | 0.177 | 0.214 | 0.172 | 0.218 | 0.169 | 0.208 | 0.170 | 0.211 | 0.169 | 0.208 | 4.1% |
| | 192 | 0.204 | 0.229 | 0.191 | 0.224 | 0.218 | 0.245 | 0.193 | 0.226 | 0.189 | 0.230 | 0.187 | 0.222 | 0.186 | 0.223 | 0.184 | 0.220 | 4.4% |
| | 336 | 0.216 | 0.239 | 0.202 | 0.234 | 0.233 | 0.258 | 0.205 | 0.238 | 0.201 | 0.240 | 0.199 | 0.232 | 0.195 | 0.233 | 0.192 | 0.227 | 4.6% |
| | 720 | 0.231 | 0.253 | 0.215 | 0.247 | 0.254 | 0.278 | 0.225 | 0.256 | 0.237 | 0.286 | 0.224 | 0.254 | 0.200 | 0.236 | 0.200 | 0.232 | 6.0% |
| PeMSD8 | 24 | 0.289 | 0.247 | 0.285 | 0.246 | 0.361 | 0.318 | 0.306 | 0.265 | 0.303 | 0.253 | 0.299 | 0.252 | 0.314 | 0.257 | 0.306 | 0.256 | 5.2% |
| | 48 | 0.367 | 0.281 | 0.356 | 0.277 | 0.475 | 0.378 | 0.386 | 0.303 | 0.342 | 0.271 | 0.340 | 0.270 | 0.357 | 0.281 | 0.356 | 0.279 | 5.8% |
| | 96 | 0.445 | 0.316 | 0.410 | 0.305 | 0.562 | 0.421 | 0.449 | 0.336 | 0.373 | 0.290 | 0.368 | 0.286 | 0.393 | 0.304 | 0.386 | 0.297 | 7.6% |
| | 192 | 0.519 | 0.354 | 0.471 | 0.337 | 0.611 | 0.443 | 0.502 | 0.364 | 0.409 | 0.312 | 0.399 | 0.303 | 0.440 | 0.333 | 0.429 | 0.324 | 7.6% |
| | 336 | 0.562 | 0.366 | 0.511 | 0.353 | 0.648 | 0.462 | 0.532 | 0.379 | 0.439 | 0.318 | 0.430 | 0.310 | 0.468 | 0.350 | 0.441 | 0.333 | 9.1% |
| | 720 | 0.653 | 0.403 | 0.563 | 0.378 | 0.748 | 0.519 | 0.597 | 0.414 | 0.488 | 0.356 | 0.468 | 0.338 | 0.511 | 0.379 | 0.484 | 0.342 | 12.3% |

PeMSD8, we conjecture that these datasets have fewer distribution shifts in channel dependence, leading to fewer overfitting risks. Even though the CD backbones have benefited from channel dependence, LIFT can still refine their predictions, *e.g.*, improving Crossformer by 4.1% on Solar. This indicates that existing CD approaches cannot fully exploit the lead-lag relationships without prior knowledge about the dynamic variation of leading indicators and leading steps. Moreover, Crossformer mixes information from the variates that show similarity at the same time step but pays insufficient attention to the different yet informative signals of leading indicators. MTGNN learns a static graph structure among variates on the training data and aggregates information within a fixed subset of variates. MTGNN may well suffer from distribution shifts in channel dependence, while LIFT dynamically selects leading indicators and reduces overfitting risks.

**LightMTS as a Strong Baseline.** Moreover, we compare the performance of LightMTS and all baselines on Weather, Electricity, and Traffic datasets. We borrow the baseline results from the paper of PatchTST with $H \in \{96, 192, 336, 720\}$. As shown in Figure 5a, LightMTS with a simple linear layer as its backbone still shows considerable performance among the state-of-the-art models. In particular, LightMTS surpasses PatchTST, the complex Transformer model, by 3.2% on Weather and 0.7% on Electricity. However, PatchTST significantly outperforms LightMTS on the Traffic dataset. As Traffic contains the greatest number of variates with complex temporal patterns, it requires a strong backbone to model the intricate cross-time dependence. Nevertheless, LightMTS is still the most competitive baseline on Traffic.

## 5.3 ABLATION STUDY

To verify the effectiveness of our designs, we introduce three variants of LightMTS by removing the influence term $\sum_{k=1}^{K} \widetilde{U}_k^{(j)}$ in Eq. (11), removing the difference term $\sum_{k=1}^{K} \widetilde{\Delta}_k^{(j)}$ in Eq. (11), and directly using $V^{(j)}$, $\sum_{k=1}^{K} U_k^{(j)}$ and $\sum_{k=1}^{K} \Delta_k^{(j)}$ in Eq. (11), respectively.

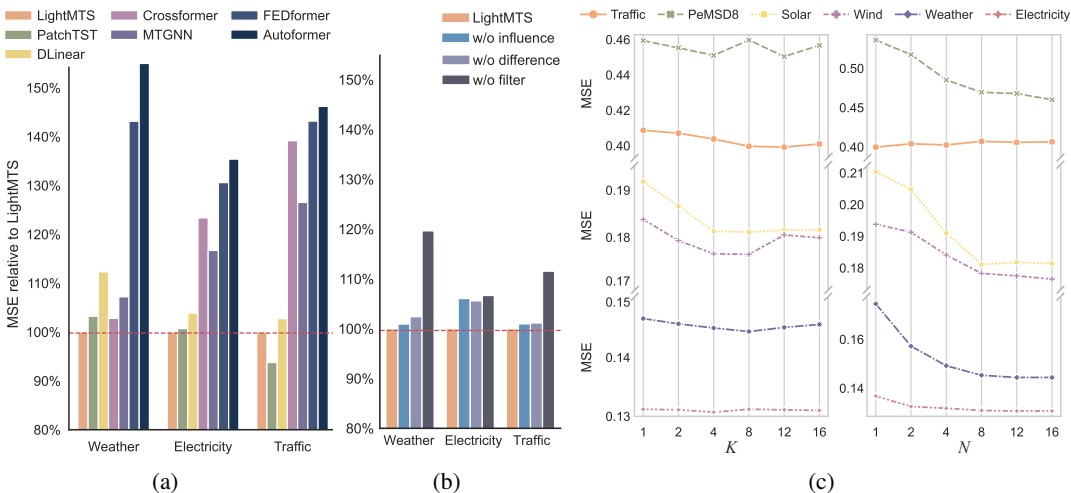

Figure 5: (a) Performance comparison between LightMTS and all baselines; (b) Performance comparison between variants of LightMTS; (c) Performance of DLinear+LIFT under different numbers of the selected leading indicators (*i.e.*, $K$) and the states (*i.e.*, $N$).

As shown in Figure 5b, we conduct experiments on these variants with $H$ set to 96, reporting the relative MSE *w.r.t.* LightMTS on Weather, Electricity, and Traffic datasets. With both the influence and the difference involved, LightMTS considers two kinds of lead-lag relationships and keeps the best performance across the datasets. In contrast, LightMTS w/o influence and LightMTS w/o difference only consider one-sided information of leading indicators, thus showing inferior performance, especially on the Electricity dataset. Furthermore, LightMTS w/o filter achieves the worst results in all the cases, which fails to adaptively filter out the noise in leading indicators.

## 5.4 HYPERPARAMETER STUDY

Our method introduces merely two additional hyperparameters, *i.e.*, the number of selected leading indicators $K$ and the number of states $N$. Thus it requires a little labor for hyperparameter selection.

With DLinear as the backbone and $H$ set to 96, we study the hyperparameter sensitivity of LIFT. As shown in Figure 5c, LIFT achieves lower MSE with an increasing $K$ on most datasets. Nevertheless, LIFT may well include more noise with a too large $K$ (*e.g.*, on the Wind dataset), resulting in performance degradation. Besides, LIFT cannot enjoy significant improvement with a larger $K$ on the Electricity dataset, where the lead-lag relationships are perhaps more sparse. As for variate states, LIFT achieves lower MSE with an increasing $N$ in most cases. We observe the most significant performance drop on Weather when ignoring the variate states. It is noteworthy that the variates of Weather (*e.g.*, wind speed, humidity, and air temperature) are recorded by various kinds of sensors, and the lead-lag patterns naturally vary with the variate states.

## 6 CONCLUSION

In this work, we rethink the channel dependence in MTS and highlight the locally stationary lead-lag relationship between variates. We propose a novel method called LIFT that efficiently estimates the relationships and dynamically incorporates leading indicators in the frequency domain for MTS forecasting. LIFT can work as a plug-and-play module and is generally applicable to arbitrary forecasting models. We further introduce LightMTS as a lightweight yet strong baseline for MTS forecasting, which keeps similar parameter efficiency to linear models and shows considerable performance. We anticipate that the lead-lag relationship can offer a novel cross-time perspective on the channel dependence in MTS, which is a promising direction for the future development of channel-dependent Transformers or other complex neural networks.

ACKNOWLEDGEMENTS

This work is supported by the National Key Research and Development Program of China (2022YFE0200500), Shanghai Municipal Science and Technology Major Project (2021SHZDZX0102), and SJTU Global Strategic Partnership Fund (2021 SJTU-HKUST).

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

# A    MATHEMATICAL PROOFS

## A.1    DISCRETE-TIME FOURIER TRANSFORM

Discrete Fourier Transform (DFT) provides a frequency-domain view of discrete time series. Given a particular frequency $f \in \{0, \frac{1}{L}, \cdots, \frac{\lfloor L/2 \rfloor}{L}\}$, the corresponding frequency component $U_f$ of univariate time series $u_{t-L+1:t}$ is derived by

$$U_f = \mathcal{F}(u_{t-L+1:t})_f = \sum_{\ell=0}^{L-1} u_{t-L+1+\ell} \cdot e^{-i2\pi f\ell}, \tag{13}$$

where $i$ is the imaginary unit and $U_f \in \mathbb{C}$ is a complex number. The inverse DFT is defined by

$$\mathcal{F}^{-1}(U)_\ell = \frac{1}{L} \sum_f U_f \cdot e^{i2\pi f\ell}. \tag{14}$$

We can calculate the amplitude $|U_f|$ and the phase $\phi(U_f)$ of the corresponding cosine signal by

$$|U_f| = \sqrt{\mathfrak{R}\{U_f\}^2 + \mathfrak{I}\{U_f\}^2} \quad \text{and} \quad \phi(U_f) = \tan^{-1}\left(\frac{\mathfrak{I}\{U_f\}}{\mathfrak{R}\{U_f\}}\right), \tag{15}$$

where $\mathfrak{R}\{U_f\}$ and $\mathfrak{I}\{U_f\}$ denotes its real and imaginary components of $U_f$. With $\mathfrak{R}\{U_f\}$ and $\mathfrak{I}\{U_f\}$ scaled in the same rate, our proposed real-valued filters only scale the amplitude but keep the phase unchanged.

## A.2    EFFICIENT CROSS-CORRELATION ESTIMATION

Given another time series $v$ that lags behind $u$ by $\delta$ steps, we denote its frequency components as $V$ and define the cross-correlation between their lookback windows by

$$R(\delta) \triangleq \frac{1}{L} \sum_{\ell=0}^{L} u_{t-L+1+\ell-\delta} \cdot v_{t-L+1+\ell} = \frac{1}{L} \sum_{\ell=0}^{L} u(\ell - \delta)v(\ell). \tag{16}$$

With $u(\ell - \delta)$ denoted as $\breve{u}(\delta - \ell)$ and $(\delta - \ell)$ denoted as $\delta'$, we derive the Fourier Transform of $\left\{\sum_{\ell=0}^{L-1} v(\ell)u(\ell - \delta)\right\}_{\delta=0}^{L-1}$ as follows.

$$\begin{aligned}
\mathcal{F}\left(\sum_{\ell=0}^{L-1} v(\ell)\breve{u}(\delta - \ell)\right)_f &= \sum_{\delta=0}^{L-1}\left(\sum_{\ell=0}^{L-1} v(\ell)\breve{u}(\delta - \ell)\right)e^{-i2\pi f\delta} \\
&= \sum_{\ell=0}^{L-1} v(\ell)\left(\sum_{\delta=0}^{L-1}\breve{u}(\delta - \ell)e^{-i2\pi f\delta}\right) \\
&= \sum_{\ell=0}^{L-1} v(\ell)e^{-i2\pi f\ell}\left(\sum_{\delta=0}^{L-1}\breve{u}(\delta - \ell)e^{-i2\pi f(\delta-\ell)}\right) \\
&= V_f\left(\sum_{\delta=0}^{L-1}\breve{u}(\delta')e^{-i2\pi f\delta'}\right).
\end{aligned} \tag{17}$$

Assuming $\check{u}$ is $L$-periodic, we have

$$
\begin{aligned}
\sum_{\delta'=-\ell}^{L-1-\ell} \check{u}(\delta') e^{-i2\pi f \delta'} &= \sum_{\delta'=-\ell}^{-1} \check{u}_{t-L+1+\delta'} \cdot e^{-i2\pi f(\delta'+L)} + \sum_{\delta'=0}^{L-1-\ell} \check{u}_{t-L+1+\delta'} \cdot e^{-i2\pi f \delta'} \\
&= \sum_{\delta'=-\ell}^{-1} \check{u}_{t-L+1+\delta'+L} \cdot e^{-i2\pi f(\delta'+L)} + \sum_{\delta'=0}^{L-1-\ell} \check{u}_{t-L+1+\delta'} \cdot e^{-i2\pi f \delta'} \\
&= \sum_{\delta'=L-\ell}^{L-1} \check{u}_{t-L+1+\delta'} \cdot e^{-i2\pi f \delta'} + \sum_{\delta'=0}^{L-1-\ell} \check{u}_{t-L+1+\delta'} \cdot e^{-i2\pi f \delta'} \\
&= \sum_{\delta'=0}^{L-1} \check{u}_{t-L+1+\delta'} \cdot e^{-i2\pi f \delta'} \\
&= \mathcal{F}(\check{u})_f.
\end{aligned}
\tag{18}
$$

Due to the conjugate symmetry of the DFT on real-valued signals, we have

$$
\mathcal{F}(\check{u})_f = \overline{\mathcal{F}(u)}_f = \overline{U_f}
\tag{19}
$$

where the bar is the conjugate operation. Thereby, we can obtain

$$
\mathcal{F}\left( \sum_{\ell=0}^{L-1} v(\ell) u(\ell - \delta) \right)_f = \mathcal{V}_f \overline{U_f},
\tag{20}
$$

Finally, we can estimate Eq. (16) as

$$
R(\delta) \approx \frac{1}{L} \mathcal{F}^{-1}(\mathcal{F}(v_{t-L+1:t}) \odot \overline{\mathcal{F}(u_{t-L+1:t})})_\delta.
\tag{21}
$$

Note that $-1 \leq R(\delta) \leq 1$ when $u_{t-L+1:t}$ and $v_{t-L+1:t}$ have been normalized.

To obtain more accurate results, one can first obtain an approximate leading step $\delta$ by Eq. (21) and Eq. (3), and then compute Eq. (1) with $\{\tau \in \mathbb{N} \mid \delta - \epsilon \leq \tau \leq \delta + \epsilon\}$, where $\epsilon \ll L$. We would like to leave this improvement as future work.

## B  DETAILS OF LEAD ESTIMATOR

Given the cross-correlation coefficients $\{R_{i,t}^{(j)}(\tau) \mid 0 \leq \tau \leq L - 1\}$ between variate $i$ and variate $j$, we identify the leading step by

$$
\delta_{i,t}^{(j)} = \arg \max_{1 \leq \tau \leq L-2} |R_{i,t}^{(j)}(\tau)|,
\tag{22}
$$

$$
\text{s.t.} \quad |R_{i,t}^{(j)}(\tau-1)| < |R_{i,t}^{(j)}(\tau)| < |R_{i,t}^{(j)}(\tau+1)|,
\tag{23}
$$

which is targeted at the globally maximal absolute cross-correlation. Note that Eq. (21) only estimates cross-correlations with $\tau$ in $\{0, \cdots, L-1\}$. If the real leading step is greater than $L - 1$ (*e.g.*, $|R_{i,t}^{(j)}(L)| > |R_{i,t}^{(j)}(L-1)|$), we could mistakenly estimate $\delta$ as $L - 1$. Therefore, we only consider the peak values as constrained by Eq. (23).

Besides, we further normalize the cross-correlation coefficients $|R_{i,t}^{(j)*}|_{i \in \mathcal{I}_t^{(j)}}$. As the evolution of the target variate is affected by both itself and the $K$ leading indicators, it is desirable to evaluate the relative leading effects. Specifically, we derive a normalized coefficient for each leading indicator $i \in \mathcal{I}_t^{(j)}$ by:

$$
\widetilde{R}_{i,t}^{(j)} = \frac{\exp |R_{i,t}^{(j)*}|}{\exp R_{j,t}^{(j)}(0) + \sum_{i' \in \mathcal{I}_t^{(j)}} \exp |R_{i',t}^{(j)*}|},
\tag{24}
$$

where $R_{j,t}^{(j)}(0) \equiv 1$. Though $\mathcal{I}_t^{(j)}$ may also involve the variate $j$ itself in periodic data, we can only include variate $j$ in its *last period* due to Eq. (23). Note that time series contains not only

the seasonality (*i.e.*, periodicity) but also its trend. Thus we use $R_{j,t}^{(j)}(0)$ to consider the *current evolution* effect from variate $j$ itself beyond its periodicity.

In terms of the proposed filter factory, we generate filters based on $\widetilde{\boldsymbol{R}}_t^{(j)} = \{\widetilde{R}_{i,t}^{(j)} \mid i \in \mathcal{I}_t^{(j)}\} \in \mathbb{R}^K$, which represents the proportion of leading effects.

## C  EXPERIMENTAL DETAILS

### C.1  DATASET DESCRIPTIONS

Table 2: The statistics of nine popular MTS datasets.

| Datasets | Weather | Electricity | Traffic | Solar | Wind | PeMSD8 | ETTm1 | ETTh1 | ILI |
|---|---|---|---|---|---|---|---|---|---|
| # of variates | 21 | 321 | 862 | 137 | 28 | 510 | 7 | 7 | 7 |
| # of timestamps | 52,696 | 26,304 | 17,544 | 52,560 | 50,000 | 17,856 | 69,680 | 17,420 | 966 |
| Sampling Rate | 10 min | 1 hour | 1 hour | 10 min | 1 hour | 5 min | 15 min | 1 hour | 1 week |

We provide the statistics of the nine popular MTS datasets in Table 2. The detailed descriptions are listed as follows.

- Electricity[3] includes the hourly electricity consumption (Kwh) of 321 clients from 2012 to 2014.

- Weather[4] includes 21 features of weather, *e.g.*, air temperature and humidity, which are recorded every 10 min for 2020 in Germany.

- Traffic[5] includes the hourly road occupancy rates recorded by the sensors of San Francisco freeways from 2015 to 2016.

- Solar[6] includes the solar power output hourly collected from 137 PV plants in Alabama State in 2007.

- Wind[7] includes hourly wind energy potential in 28 European countries. We collect the latest 50,000 records (about six years) before 2015.

- PeMSD8[8] includes the traffic flow, occupation, and speed in San Bernardino from July to August in 2016, which are recorded every 5 min by 170 detectors. We take the dataset as an MTS of 510 channels in most experiments, while only MTGNN models the 170 detectors with three features for each detector.

- ETT (Electricity Transformer Temperature)[9] includes seven oil and load features of electricity transformers from July 2016 to July 2018. ETTm1 is 15-minutely collected and ETTh1 is hourly collected.

- ILI [10] includes the ratio of patients seen with influenzalike illness and the number of patients. It includes weekly data from the Centers for Disease Control and Prevention of the United States from 2002 to 2021.

To evaluate the forecasting performance of the baselines, we divide each dataset into the training set, validation set, and test set by the ratio of 7:1:2.

---

[3] https://archive.ics.uci.edu/dataset/321/electricityloaddiagrams20112014/
[4] https://www.bgc-jena.mpg.de/wetter/
[5] http://pems.dot.ca.gov/
[6] https://www.nrel.gov/grid/solar-power-data/
[7] https://www.kaggle.com/datasets/sohier/30-years-of-european-wind-generation/
[8] https://github.com/wanhuaiyu/ASTGCN/
[9] https://github.com/zhouhaoyi/ETDataset
[10] https://gis.cdc.gov/grasp/fluview/fluportaldashboard.html

Table 3: Comparison of practical efficiency of PatchTST, Crossformer, and LIFT with $H = 720$. MACs are the number of multiply-accumulate operations per sample. The batch size is set to 1.

| | | PatchTST | +LIFT | Relative additional cost | Crossformer | +LIFT | Relative additional cost |
|---|---|---|---|---|---|---|---|
| Weather | Parameter (M) | 4.3 | 4.7 | 9.3% | 11.7 | 12.5 | 7.1% |
| | MACs (G) | 0.5 | 0.5 | 1.8% | 4.4 | 4.4 | 0.2% |
| | Memory (MB) | 43 | 46 | 7.7% | 206 | 209 | 1.6% |
| | Time (ms) | 4.7 | 6.0 | 29.6% | 37.7 | 40.6 | 7.6% |
| Electricity | Parameter (M) | 4.3 | 4.7 | 9.3% | 2.4 | 2.8 | 16.7% |
| | MACs (G) | 7.1 | 7.2 | 1.8% | 8.6 | 8.7 | 1.4% |
| | Memory (MB) | 0.4 | 0.4 | 1.2% | 873 | 878 | 0.6% |
| | Time (ms) | 5.1 | 6.0 | 17.4% | 31.5 | 33.1 | 5.1% |
| Traffic | Parameter (M) | 4.3 | 4.7 | 9.3% | 3.2 | 3.6 | 12.4% |
| | MACs (G) | 19.0 | 19.4 | 1.8% | 10.6 | 10.9 | 3.2% |
| | Memory (MB) | 1061 | 1092 | 2.9% | 1558 | 1578 | 1.3% |
| | Time (ms) | 5.1 | 5.8 | 14.5% | 32.7 | 36.4 | 11.3% |
| Solar | Parameter (M) | 2.0 | 2.9 | 41.2% | 1.9 | 2.3 | 23.4% |
| | MACs (G) | 0.9 | 1.0 | 5.3% | 3.7 | 3.7 | 1.6% |
| | Memory (MB) | 92 | 96 | 4.3% | 377 | 381 | 1.0% |
| | Time (ms) | 4.9 | 6.0 | 23.4% | 31.9 | 33.4 | 5.0% |
| Wind | Parameter (M) | 2.0 | 2.5 | 20.6% | 11.8 | 12.2 | 3.4% |
| | MACs (G) | 0.2 | 0.2 | 5.6% | 5.7 | 5.8 | 0.2% |
| | Memory (MB) | 25 | 29 | 13.1% | 256 | 259 | 1.3% |
| | Time (ms) | 4.7 | 6.1 | 28.9% | 45.2 | 47.2 | 4.4% |
| PeMSD8 | Parameter (M) | 2.0 | 2.8 | 35.8% | 2.9 | 4.0 | 36.5% |
| | MACs (G) | 3.5 | 3.7 | 5.6% | 13.6 | 13.8 | 1.5% |
| | Memory (MB) | 319 | 325 | 1.9% | 1381 | 1387 | 0.4% |
| | Time (ms) | 5.6 | 6.0 | 7.0% | 32.4 | 33.7 | 4.3% |

## C.2 IMPLEMENTATION DETAILS

All experiments are conducted on a single Nvidia A100 40GB GPU. We use the official implementations of all baselines and follow their recommended hyperparameters. Typically, the batch size is set to 32 for most baselines, while PatchTST recommends 128 for Weather. We adopt the Adam optimizer and search the optimal learning rate in {0.5, 0.1, 0.05, 0.01, 0.005, 0.001, 0.0005, 0.0001, 0.00005, 0.00001}. As for LIFT, we continue the grid search with $K$ in {1, 2, 4, 8, 12, 16} and $N$ in {1, 2, 4, 8, 12, 16}. Note that we stop the hyperparameter tuning for consistent performance drop along one dimension of the hyperparameters, *i.e.*, we only conduct the search in a subset of the grid.

As the Lead-aware Refiner has a few dependencies on the preliminary predictions from the backbone, it is sometimes hard to train LIFT in the early epochs, especially when using complex CD backbones. To speed up the convergence, one alternative way is to pretrain the backbone for epochs and then jointly train the framework. In our experiments, we report the best result in Table 2, while DLinear+LIFT and LightMTS are always trained in an end-to-end manner.

## D ADDITIONAL EXPERIMENTS

## D.1 EFFICIENCY STUDY

Following Zeng et al. (2023), we compare the backbones and LIFT by the number of parameters, the number of operations, the GPU memory consumption, and the inference time. As shown in Table 3, LIFT additionally requires an average of 10.7% parameters, 1.7% MACs, 2.5% GPU memory, and 14.2% inference time more than its backbone. It is also noteworthy that our Lead Estimator is non-parametric and can pre-compute estimation only once on the training data, reducing the practical training time.

Moreover, we compare the parameter efficiency of LightMTS and all baselines on the six datasets. As shown in Figure 6, LightMTS keeps similar parameter efficiency with DLinear, a simple linear

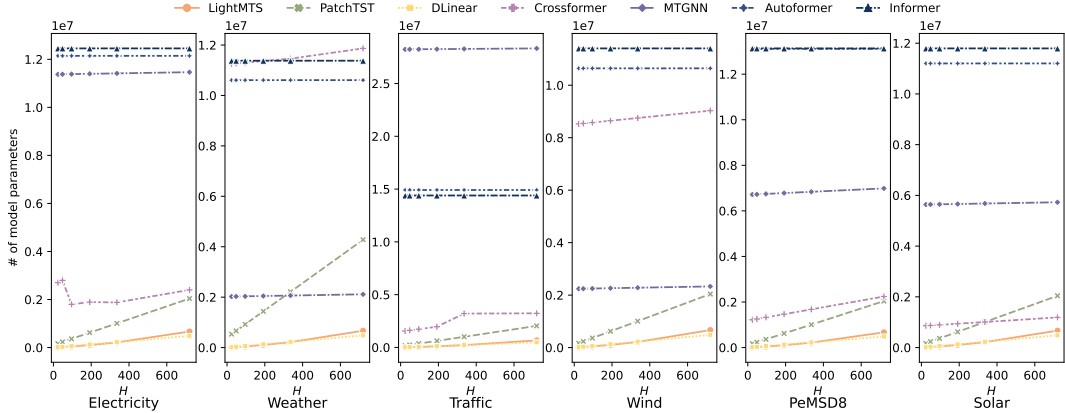

Figure 6: The number of model parameters on six datasets with horizon $H$ in $\{24, 48, 96, 192, 336, 720\}$. For Crossformer, we follow its recommended lookback length $L$. For Informer and Autoformer, $L$ is 96. For other methods, $L$ is 336.

model. On average, the parameter size of LightMTS is 1/5 of PatchTST, 1/25 of Crossformer, 1/50 of MTGNN, and 1/70 of Informer and Autoformer. It is noteworthy that a larger $H$ enlarges the gap between PatchTST and LightMTS because PatchTST employs a fully connected layer to decode the $H$-length sequence of high-dimensional hidden states. Although the parameter sizes of Informer and Autoformer are irrelevant to $H$, they are still the most parameter-heavy due to their high-dimensional learning throughout encoding and decoding.

## D.2 DISTRIBUTION SHIFTS

To investigate the dynamic variation of lead-lag relationships, we adopt the proposed lead estimator ($K = 2$) to count the lead-lag relationships in the training data and test data. As shown in Figure 7, some leading indicators of a specific variate in the training data cannot keep the relationships in the test data, while the test data also encounters new patterns. GNN-based (Wu et al., 2020) and MLP-based (Li et al., 2023b) methods are susceptible to such distribution shifts of leading indicators due to their static parameter weights in modeling channel dependence.

Furthermore, we visualize the distribution of leading steps between the pair of variates. We choose a lagged variate and its leading indicator that is the most commonly observed across training and test data. As shown in Figure 8, some of the leading steps (*e.g.*, 250) observed in training data rarely reoccur in the test data. By contrast, the leading indicator show new leading steps (*e.g.*, 40 and 125)

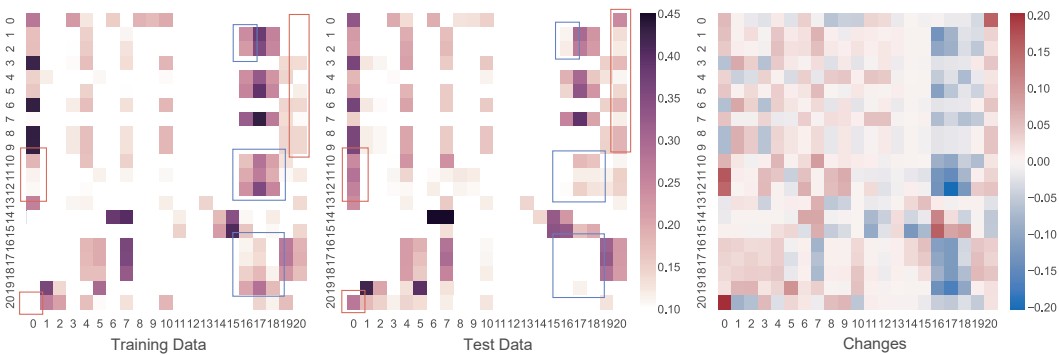

Figure 7: Distributions of leading indicators ($K = 2$) in the training data (*left*) and test data (*mid*) on the Weather dataset, where each cell represents the occurrence frequency of the lead-lag relationship between each pair of variates. The *right* shows the changes in occurrence frequency from training data to test data.

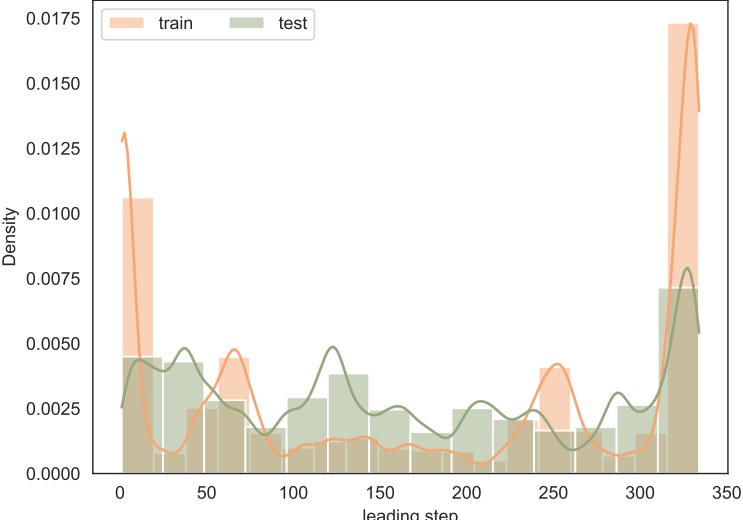

Figure 8: The histogram of the leading step between a selected pair of variates in the training data and test data on the Weather dataset. We also estimate the distributions with a kernel density estimator.

Table 4: Performance comparison in terms of forecasting errors. We highlight the better results between each pair of backbones and LIFT in **bold**.

| Method | | PatchTST | | + LIFT | | DLinear | | + LIFT | |
|---|---|---|---|---|---|---|---|---|---|
| | | MSE | MAE | MSE | MAE | MSE | MAE | MSE | MAE |
| ETTh1 | 24 | 0.307 | 0.358 | **0.303** | **0.356** | 0.323 | 0.367 | **0.316** | **0.360** |
| | 48 | 0.338 | 0.374 | **0.337** | 0.374 | 0.345 | 0.378 | **0.342** | **0.376** |
| | 96 | 0.375 | 0.399 | **0.370** | **0.395** | 0.375 | 0.399 | **0.372** | **0.395** |
| | 192 | 0.414 | 0.421 | **0.410** | **0.419** | **0.405** | **0.416** | 0.413 | 0.423 |
| | 336 | 0.431 | 0.436 | 0.433 | **0.435** | **0.439** | **0.443** | 0.453 | 0.453 |
| | 720 | 0.449 | 0.466 | **0.447** | **0.464** | **0.472** | **0.490** | 0.509 | 0.512 |
| ETTm1 | 24 | 0.193 | 0.270 | **0.190** | **0.269** | 0.211 | 0.285 | **0.196** | **0.275** |
| | 48 | 0.254 | 0.319 | **0.252** | **0.316** | 0.272 | 0.326 | **0.258** | **0.321** |
| | 96 | 0.290 | 0.342 | **0.287** | **0.338** | 0.299 | 0.343 | **0.293** | **0.342** |
| | 192 | 0.332 | 0.369 | **0.329** | **0.367** | 0.335 | 0.365 | **0.334** | 0.365 |
| | 336 | 0.366 | 0.392 | **0.365** | **0.390** | 0.369 | **0.386** | 0.369 | 0.387 |
| | 720 | 0.420 | 0.424 | **0.412** | **0.420** | **0.425** | **0.421** | 0.426 | 0.424 |
| ILI | 6 | 0.909 | 0.562 | **0.819** | **0.543** | 1.115 | 0.713 | **0.974** | **0.645** |
| | 12 | 1.523 | 0.764 | **1.378** | **0.726** | 1.844 | 0.941 | **1.705** | **0.904** |
| | 24 | 1.767 | **0.830** | **1.677** | 0.841 | 2.215 | 1.081 | **2.083** | **1.002** |
| | 36 | 1.651 | 0.857 | **1.629** | **0.830** | 2.301 | 1.076 | **2.172** | **1.033** |
| | 48 | 1.711 | 0.859 | **1.668** | **0.844** | 2.316 | 1.091 | **2.187** | **1.052** |
| | 60 | 1.816 | 0.884 | **1.770** | **0.880** | 2.445 | 1.123 | **2.342** | **1.094** |

in the test data. Furthermore, the leading step is not fixed but dynamically varies across phases, increasing the difficulty of modeling channel dependence.

## D.3 PERFORMANCE ON OTHER DATASETS

We conduct more experiments on Illness, ETTm1, and ETTh1 datasets. As PatchTST and DLinear perform the best on these benchmarks, we employ them as backbones. As shown in Table 4, LIFT fails in some cases. We reason that these datasets are composed of only 7 variates and perhaps have insufficient leading indicators for each variate. Nevertheless, it is worth mentioning that we can collect abundant variates in practical applications, paving the way for adopting LIFT.

Table 5: Performance comparison in terms of forecasting errors. The best results are in **bold** and the second best are underlined

| Models | | LightMTS | | PatchTST | | Dlinear | | Crossformer | | MTGNN | | FEDformer | | Autoformer | | Informer | |
|---|---|---|---|---|---|---|---|---|---|---|---|---|---|---|---|---|---|
| Metric | | MSE | MAE | MSE | MAE | MSE | MAE | MSE | MAE | MSE | MAE | MSE | MAE | MSE | MAE | MSE | MAE |
| Weather | 96 | 0.147 | **0.198** | 0.152 | 0.199 | 0.176 | 0.237 | **0.145** | 0.209 | 0.157 | 0.216 | 0.238 | 0.314 | 0.249 | 0.329 | 0.354 | 0.405 |
| | 192 | **0.190** | **0.238** | 0.197 | 0.243 | 0.220 | 0.282 | 0.197 | 0.264 | 0.205 | 0.269 | 0.275 | 0.329 | 0.325 | 0.370 | 0.419 | 0.434 |
| | 336 | **0.240** | **0.278** | 0.249 | 0.283 | 0.265 | 0.319 | 0.246 | 0.309 | 0.258 | 0.312 | 0.339 | 0.377 | 0.351 | 0.391 | 0.583 | 0.543 |
| | 720 | **0.318** | **0.332** | 0.320 | 0.335 | 0.323 | 0.362 | 0.323 | 0.364 | 0.335 | 0.369 | 0.389 | 0.409 | 0.415 | 0.426 | 0.916 | 0.705 |
| Electricity | 96 | 0.131 | 0.224 | **0.130** | **0.222** | 0.140 | 0.237 | 0.142 | 0.243 | 0.139 | 0.242 | 0.186 | 0.302 | 0.196 | 0.313 | 0.304 | 0.393 |
| | 192 | **0.148** | **0.240** | **0.148** | **0.240** | 0.153 | 0.249 | 0.159 | 0.259 | 0.160 | 0.261 | 0.197 | 0.311 | 0.211 | 0.324 | 0.327 | 0.417 |
| | 336 | **0.165** | **0.257** | 0.167 | 0.261 | 0.169 | 0.267 | 0.192 | 0.293 | 0.193 | 0.284 | 0.213 | 0.328 | 0.214 | 0.327 | 0.333 | 0.422 |
| | 720 | 0.203 | **0.288** | **0.202** | 0.291 | 0.203 | 0.301 | 0.225 | 0.316 | 0.242 | 0.327 | 0.233 | 0.344 | 0.236 | 0.342 | 0.351 | 0.427 |
| Traffic | 96 | 0.386 | 0.268 | **0.367** | **0.251** | 0.410 | 0.282 | 0.519 | 0.293 | 0.479 | 0.289 | 0.576 | 0.359 | 0.597 | 0.371 | 0.733 | 0.410 |
| | 192 | 0.405 | 0.276 | **0.385** | **0.259** | 0.423 | 0.287 | 0.522 | 0.296 | 0.507 | 0.307 | 0.610 | 0.380 | 0.607 | 0.382 | 0.777 | 0.435 |
| | 336 | 0.417 | 0.282 | **0.398** | **0.265** | 0.436 | 0.296 | 0.530 | 0.300 | 0.539 | 0.314 | 0.608 | 0.375 | 0.623 | 0.387 | 0.776 | 0.434 |
| | 720 | 0.444 | 0.298 | **0.434** | **0.287** | 0.466 | 0.315 | 0.584 | 0.369 | 0.616 | 0.352 | 0.621 | 0.375 | 0.639 | 0.395 | 0.827 | 0.466 |
| ETTh1 | 96 | 0.311 | 0.359 | **0.307** | **0.358** | 0.375 | 0.399 | 0.423 | 0.448 | 0.440 | 0.450 | 0.326 | 0.390 | 0.510 | 0.492 | 0.626 | 0.560 |
| | 192 | 0.341 | **0.374** | **0.338** | **0.374** | 0.405 | 0.416 | 0.471 | 0.474 | 0.449 | 0.433 | 0.365 | 0.415 | 0.514 | 0.495 | 0.725 | 0.619 |
| | 336 | **0.369** | **0.391** | 0.375 | 0.399 | 0.439 | 0.443 | 0.570 | 0.546 | 0.598 | 0.554 | 0.392 | 0.425 | 0.510 | 0.492 | 1.005 | 0.741 |
| | 720 | **0.407** | **0.416** | 0.414 | 0.421 | 0.472 | 0.490 | 0.653 | 0.621 | 0.685 | 0.620 | 0.446 | 0.458 | 0.527 | 0.493 | 1.133 | 0.845 |
| ETTm1 | 96 | **0.285** | **0.340** | 0.290 | 0.342 | 0.299 | 0.343 | 0.315 | 0.370 | 0.330 | 0.388 | 0.326 | 0.390 | 0.510 | 0.492 | 0.626 | 0.560 |
| | 192 | **0.325** | **0.364** | 0.332 | 0.365 | 0.335 | 0.365 | 0.348 | 0.390 | 0.376 | 0.419 | 0.365 | 0.415 | 0.514 | 0.495 | 0.725 | 0.619 |
| | 336 | **0.365** | **0.384** | 0.366 | 0.392 | 0.369 | 0.386 | 0.414 | 0.432 | 0.432 | 0.461 | 0.392 | 0.425 | 0.510 | 0.492 | 1.005 | 0.741 |
| | 720 | 0.422 | 0.423 | **0.420** | 0.424 | 0.425 | **0.421** | 0.511 | 0.552 | 0.485 | 0.488 | 0.446 | 0.458 | 0.527 | 0.493 | 1.133 | 0.845 |

## D.4 PERFORMANCE OF LIGHTMTS

In Table 5, we compare the forecasting errors of LightMTS with all baselines. The lookback length $L$ is set to 336 for LightMTS, PatchTST, DLinear, Crossformer, and MTGNN. We borrow the results of other baselines from the paper of PatchTST. As shown in Table 5, LightMTS achieves comparable performance in all the cases.

