# OpenReview forum: "Rethinking Channel Dependence for Multivariate Time Series Forecasting: Learning from Leading Indicators"
_ICLR.cc/2024/Conference — ICLR 2024 poster_

### Official Review · Reviewer_4A7m · 2023-11-08

**Soundness:** 3 good
**Presentation:** 3 good
**Contribution:** 3 good
**Rating:** 6
**Confidence:** 3

**Summary:**

The paper introduces LIFT, a novel method aimed at enhancing multivariate time series (MTS) forecasting by leveraging locally stationary lead-lag relationships between variates. The authors suggest that recognizing and utilizing the dependence between channels can significantly improve forecasting accuracy. LIFT is designed to be a flexible plug-in that can integrate with any existing time series forecasting methods. The method was tested across several real-world datasets, showing remarkable improvement. Additionally, the paper presents LightMTS, a new baseline model for MTS forecasting that maintains parameter efficiency while offering competitive performance.

**Strengths:**

**Originality** The paper proposes a novel approach to leverage the lead-lag relation to contribute to MTS forecasting.

**Quality** The experiment is solid and improvement is fruitful.

**Significance** The proposed framework is flexible and can be incorporated with time series forecasting backbones, indicating its potential to improve predictive performance in a myriad of applications that depend on time series analysis.

**Clarity** The paper is well-written and easy to follow.

**Weaknesses:**

To me, the paper is well written, logically fluent and the experiment result is fruitful. My only concern is about potential contradiction in results and the slightly limited applicability. Please see the Question part for more details.

**Questions:**

Please allow me to preface my questions by mentioning that I am relatively new to this field, and as such, my inquiries may not reflect a deep understanding of the intricate concepts presented.

The following question is my major concern:

Q1. While the motivation lies in the exploitation of dependence between variables, the experiment shows better performance and improvement for CI methods compared with CD methods, which is kind of contradicted. Could you provide more insight into the possible reasons?

&nbsp;

The following questions stem from what I believe are confusions arising from differences between different fields (which won't influence my rating):

Q2.  Is it possible to provide a formal definition of  **locally stationary lead-lag relation** as "influence requires a certain time delay to propagate and take effect" is not clear to interpret mathematically? Furthermore, in the context of cross-correlation used to quantify such relationships, does this indicate that a lead-lag relationship is inferred whenever we observe relatively larger cross-correlation in differing time slots between two variables? (In the causality field, the correlation is not equivalent to influence and effect).

Q3. As a continuation of my previous question, the use of cross-correlation for identifying lead-lag relationships suggests a focus on linear associations. May I inquire if this suggests that the algorithm's applicability is confined to variables that share a linear relationship (e.g., $X_1 = X_2^2$, the cross-correlation will return zero)?

---

> ### Author Response · Authors · 2023-11-15
> **Response to Reviewer 4A7m**
>
> Thanks for your time and effort in reviewing our paper! We have provided more explanations below to better clarify our work.
>
> **Q1: The experiment shows better performance and improvement for CI methods compared with CD methods. Could you provide possible reasons?**
>
> **A1:** Let "$>$" mean "outperforms". The following is the rationale for the experimental results.
>
> $\text{Why CI backbones} > \text{CD backbones}?$
>
> Crossformer focuses on variables that share similarities at the same time but ignores leading indicators that look different at the same time (as illustrated in Figure 1) due to a large leading step. MTGNN learns a fixed set of indicators and overlooks the varying leading steps. Thus MTGNN can easily overfit training data, of which the lead-lag relationships are different from test data (see Figure 7-8). Other CD backbones (*e.g.*, Autoformer and FEDformer) simply mix up all variables and suffer from noise.
>
> $\text{Why CI+LIFT}>\text{CI} > \text{CD+LIFT}?$
>
> This happens when $CI \gg CD$ and the performance gap is too large for LIFT to fill. The reason is that LIFT is affected by the error from the backbone. As defined in Eq. (5), the shifted window of leading indicators is padded by a part of the backbone's forecasts when the leading step is smaller than the forecast horizon.
>
> $\text{Why LIFT achieves higher relative improvement on CI than CD}?$
>
> As both CD backbones and the LIFT module exploit dependence between variables, their leveraged dependence information can overlap. By contrast, CI backbones treat each variate independently and hence benefit more from the channel dependence exploited by LIFT.
>
> ----
>
> **Q2: Provide a formal definition of locally stationary lead-lag relation. Is a lead-lag relationship inferred whenever we observe relatively larger cross-correlation?**
>
> **A2:** Formally, there is a locally stationary lead-lag relationship between variate $i$ and $j$ at time $t$ if
> \\begin{aligned}
> \exists \delta >0,\\ &D(\mathcal X^{(j)} _{t+1:t+H}, \mathcal X^{(i)} _{t+1-\delta:t+H-\delta})\approx D(\mathcal X^{(j)} _{t-L+1:t}, \mathcal X^{(i)} _{t-L+1-\delta:t-\delta})\approx 0 \\\\
> \wedge\exists t'> t,\\ &D(\mathcal X^{(j)} _{t'+1:t'+H}, \mathcal X^{(i)} _{t'+1-\delta:t'+H-\delta}) \not\approx 0,
> \\end{aligned}
> where $D(\mathcal W,\mathcal W')$ is the distance between two windows $\mathcal W$ and $\mathcal W'$, and $D(\cdot,\cdot)$ can be estimated by a neural network or a non-parametric metric (*e.g.*, $1-\text{cross-correlation coefficient}$). Furthermore, if there exists $i\neq j$ such that $D(\mathcal X^{(j)} _{t+1:t+H}, \mathcal X^{(i)} _{t+1:t+H})\approx 0$, we dismiss variate $i$ as it is not a leading indicator and offers no advance information.
>
> Since most time series are not extremely non-stationary, we assume the lead-lag relationship keeps stationary across the lookback and horizon windows, *i.e.*, $\mathcal X^{(j)} _{t-L+1:t}$ and $\mathcal X^{(j)} _{t+1:t+H}$ has the same leading indicator $i$ with the leading step close to $\delta$.
>
> A lead-lag relationship is not inferred by the correlation-based estimation alone. The motivation of using cross-correlation is to estimate all variates *as fast as possible* and filter out most noisy variates. A large correlation between two lookback windows just suggests that the indicator can serve as a potential reference for the horizon window. We rely on the Lead-aware Refiner which is the key component to exploit the advance information from potential leading indicators. The Lead-aware Refiner is trained to learn how much we can trust the suggestion and how we leverage the potential leading indicators.
>
> ----
>
> **Q3: Is the algorithm's applicability confined to variables that share a linear relationship?**
>
> **A3:** We do not constrain the relationship to be linear. We just use cross-correlations to roughly yet quickly select a subset of variates. Then, our Adaptive Frequency Mixer, a non-linear neural network, is capable of modeling complex relationships within the small subset, instead of costly modeling relationships among all variates.
>
> Besides, the correlation-based estimation also allows some non-linear relationships between two windows $\mathcal W$ and $\mathcal W'$, *e.g.*, $\mathcal W=(\mathcal W')^3$, as long as the variates share similar trends. In case the subset overlooks arbitrarily complex relationships, *e.g.*, $\mathcal W=(\mathcal W')^2$, an alternative implementation of the distance function $D(\cdot, \cdot)$ is using another neural network (*e.g.*, $D(\mathcal W, \mathcal W')=f _{1}(\mathcal W)^\top f _2(\mathcal W')$, where $f _{1}$ and $f _{2}$ are two different MLPs and trained by gradient descent).

---

> ### Author Response · Authors · 2023-11-21
> **Looking Forward to Your Feedback**
>
> Dear Reviewer 4A7m,
>
> Thank you again for your dedication to reviewing our paper! As the discussion period is coming to an end very soon, we are looking forward to any feedback you have to our response. We will highly appreciate it if you let us know whether your previous concerns have been adequately addressed.

---

> > ### Comment · Reviewer_4A7m · 2023-11-21
> > **Quick response to the author**
> >
> > I appreciate the author's rebuttal and their response to my question. I will maintain my current score.

---

### Official Review · Reviewer_cNYF · 2023-11-09

**Soundness:** 3 good
**Presentation:** 3 good
**Contribution:** 2 fair
**Rating:** 6
**Confidence:** 2

**Summary:**

This paper presented LIFT, a post-hoc plugin that enhance performance for time series forecasting tasks.

LIFT works on top of black-box backbone forecasting models. It first estimates leading indicators and then shift those indicators to sync with target variate. Finally, filter based Adaptive frequency mixer was used to extract valuable information from leading indicators.

Extensive experiments on six real-world datasets demonstrate the effectiveness of LIFT.

**Strengths:**

1. The design to sync the series with leading indicators is intuitive.
2. Clear design of LIFT starting from preliminary forecasting to lead estimation and target-oriented shifts to reconstruction for the prediction.
3. Very smart post-hoc process which in orthogonal to existing time series models.
4. Proposed LightMTS a simple yet effective lightweight baseline method for MTS forecasting.

**Weaknesses:**

1. The proposed correlation based leading indicators discovery method may over-simplify the dynamics in the time series data. If the time-delayed relation among those variates is highly complex which cannot easily be captured by cross-correlation coefficient, the estimation of the leading indicators may not be accurate. Some discussion on this case is expected.
2. As introduced in locally stationary lead-lag relationship, such relationship should be static only in a short period of time, which may indicate that it may not work well in long-term forecasting. However, the author also claims good performance for that case. Some elaboration on this will be very helpful.
3. As a plug-and-play module that works on top of the baseline predictions, some fair comparison about the computation cost such as the total computation load should be considered. Otherwise, it is not surprised to see that after injecting some inductive bias, the performance is better than baselines.

**Questions:**

See Weaknesses.

---

> ### Author Response · Authors · 2023-11-16
> **Response to Reviewer cNYF**
>
> Thanks for your time and valuable comments! We are happy to see that you found our design intuitive and smart. We have provided more explanations and experiments to address your concern.
>
> **Q1: The proposed estimation of leading indicators cannot easily capture highly complex relations among variates and may not be accurate.**
>
> **A1:** Our procedure is to quickly filter out most noisy variates, coarsely obtain advance information from potential leading indicators, and then finely model relationships within a narrowed scope that benefits the forecasting. The motivation is that many MTS contain numerous variates (e.g., the Traffic dataset has 862 variates), while the complexity of estimating all cross-variate relationships at one time step is $\mathcal O(C^2)$ ($C$ is the number of variates). This requires making the estimation *as efficient as possible*, since we need to restart estimations at each prediction time.
>
> It is not necessary for our lead estimation to be perfect, because our proposed Adaptive Frequency Mixer, a non-linear neural network, shoulders the responsibility for filtering out noisy information from the selected indicators.
>
> Existing MTS forecasting methods pay little attention to the dynamics of leading steps, making their test performance susceptible to variations of channel dependence. Going a step further, one of our contributions is that we propose target-oriented shifts to keep variates aligned dynamically, allowing test-time adaptation.
>
> ---
>
> **Q2: The locally stationary lead-lag relationship should be static only in a short period, and may not work well in long-term forecasting.**
>
> **A2:** Nice comment! We provide additional data analyses on long-term forecasting benchmarks, demonstrating that the discovered lead-lag relationships generally last over the large horizon window with $H$ steps, where $H$ is set to $720$. Empirically, at each time $t$, we lookback the previous $336$ steps to select the $4$ most possible leading indicators for each target variate $j$. And for each leading indicator $i$, we obtain its leading step $\delta^{(j)} _{i,t}$. Then, we compute the future cross-correlation coefficient $R^{(j)} _{i,t+H}$ between *the ground-truth horizon window* $\mathcal X^{(j)} _{t+1:t+H}$ and the shifted sequence $\mathcal X^{(i)} _{t+1-\delta^{(j)} _{i,t}:t+H-\delta^{(j)} _{i,t}}$. With the horizon window sliding along the dataset, we repeat discovering relationships on the corresponding lookback window and computing $R^{(j)} _{i,t+H}$ for each $j$ and $i$ at different time $t$. We report the average of $R^{(j)} _{i,t+H}$ on each dataset as listed below.
>
> |Weather|Electricity|Traffic|Solar|Wind|PeMSD8|
> |--|--|--|--|--|--|
> |0.82|0.82|0.79|0.87|0.79|0.54|
>
> As shown in the table, the large values indicate the lead-lag relationships generally hold over the large horizon window in long-term forecasting tasks.
>
> There are two rationales behind the empirical results and the good performance of LIFT. (1) As most time series are not extremely non-stationary, there are relationships that last over a certain period in favor of channel dependence modeling. (2) Existing long-term forecasting methods lack prior knowledge of channel dependence and try to learn arbitrarily complex dependence, increasing difficulty in optimization and risking overfitting issues. By contrast, our method dynamically learns from the discovered lead-lag relationships, which are much simpler and remain effective during test time.
>
> ---
>
> **Q3: Fair comparison of the computation cost.**
>
> **A3:** Following LSTF-Linear [1], we compare the backbones and LIFT per sample by the number of parameters, the number of multiply-accumulate operations (MACs, $\approx\frac{1}{2}$FLOPs), the GPU memory consumption, and the inference time in the following table.
>
> |||PatchTST|PatchTST+LIFT|relative additional cost|Crossformer|Crossformer+LIFT|relative additional cost|
> |--|--|--|--|--|--|--|--|
> |**Weather**|**Parameter** (M)|4.3|4.7|9.3%|11.7|12.5|7.1%|
> ||**MACs** (G)|0.5|0.5|1.8%|4.4|4.4|0.2%|
> ||**Memory** (MB)|43|46|7.7%|206|209|1.6%|
> ||**Time** (ms)|4.7|6.0|29.6%|37.7|40.6|7.6%|
> |**Electricity**|**Parameter** (M)|4.3|4.7|9.3%|2.4|2.8|16.7%|
> ||**MACs** (G)|7.1|7.2|1.8%|8.6|8.7|1.4%|
> ||**Memory** (MB)|406|411|1.2%|873|878|0.6%|
> ||**Time** (ms)|5.1|6.0|17.4%|31.5|33.1|5.1%|
> |**Traffic**|**Parameter** (M)|4.3|4.7|9.3%|3.2|3.6|12.4%|
> ||**MACs** (G)|19.0|19.4|1.8%|10.6|10.9|3.2%|
> ||**Memory** (MB)|1061|1092|2.9%|1558|1578|1.3%|
> ||**Time** (ms)|5.1|5.8|14.5%|32.7|36.4|11.3%|
>
> Our empirical study shows that LIFT additionally requires an average of 10.7% parameters, 1.7% MACs, 2.5% memory, and 14.2% inference time more than its backbone. It is noteworthy that our Lead Estimator is non-parametric and can precompute estimation only once on the training data, reducing practical training time.
>
> ---
> [1] Zeng & Chen et. al. Are Transformers Effective for Time Series Forecasting? AAAI 2023.

---

> ### Author Response · Authors · 2023-11-21
> **Looking Forward to Your Feedback**
>
> Dear Reviewer cNYF,
>
> Thank you once again for your dedication to reviewing our paper.
>
> In our early response, we have included more explanations about our method, along with detailed data analyses and cost studies.
>
> We would like to know whether our response has addressed your concerns. If you have any further questions, please do not hesitate to let us know, so that we can respond to them timely.

---

> > ### Comment · Reviewer_cNYF · 2023-11-23
> >
> > I appreciate the author's rebuttal and their response to my question. I have increased my score.

---

### Official Review · Reviewer_MEf1 · 2023-11-09

**Soundness:** 3 good
**Presentation:** 4 excellent
**Contribution:** 2 fair
**Rating:** 6
**Confidence:** 3

**Summary:**

This paper considers the problem of multivariate time series forecasting. The authors propose a forecasting algorithm called LIFT that exploits dependencies among the time series by first finding a set of leading time series and their corresponding leading steps for a given time series and then uses a Lead-aware Refiner that adaptively leverages the informative signals of leading indicators in the frequency domain to refine the predictions of lagged variates. The first step, i.e., finding the leading time series and their corresponding steps is designed based on Wiener–Khinchin theorem that uses fast Fourier transformation. The second step contains two main parts: state estimation and a frequency mixer. They evaluate the performance of LIFT through several experiments and their result shows that LIFT makes an average improvement over both CI and CD methods.

**Strengths:**

They study an important problem. The paper is well-written.

Using interdependencies among the time series to better forecast is more intuitive than CI methods and this is showing by this work empirically.

Based on the presented evidence in empirical study, the proposed method shows improvement compared to the state of the art.

**Weaknesses:**

The main concern is the generality of the proposed method to capture general forms of interactions among time series. The leading relationship in this work is detected based on pairwise cross-correlation. What if there are more complex interactions in which two or more time series jointly influence another time series? A reason that in some applications CI is outperforming CD methods could be the result of such misspecified interactions in the CD methods.

Consider a setting in which the influence structure among  the time series is a chain, i.e., $X^{(1)} -> X^{(2)} -> \cdots -> X^{(C)}$. In this case, there are scenarios in which the pairwise cross-correlations between $X^{(C)}$ and {$X^{(i)}: 1\leq i\leq C-2$} are all higher than the correlation between $X^{(C)}$ and $X^{(C-1)}$. This means that LIFT will not pick  $X^{(C-1)}$ for forecasting while the only relevant time series for forecasting $X^{(C)}$ is $X^{(C-1)}$.

**Questions:**

Is it possible to discover causal relationships (e.g., Granger causality) among time series via modifications of LIFT?


Why there is subscript $\Delta$ and $U$ in equation (9)?

---

> ### Author Response · Authors · 2023-11-15
> **Response to Reviewer MEf1**
>
> Thanks for your time and interesting questions. We have provided more explanations below to better clarify our work and address your concern.
>
> **Q1: What if two or more time series jointly influence another time series? Misspecifying such actions interaction could be why CD methods underperform CI.**
>
> **A1:** To handle the complex cases, we design the Lead-aware Refiner to jointly model all the selected leading indicators for a target variate. Specifically, in Eq. (9), we utilize $\boldsymbol{R} _t^{(j)} \in \mathbb{R}^K$, the $K$-dimensional correlation vector of the $K$ leading indicators, to generate filters that reweight the indicators' influence. In this way, we learn from each leading indicator according to the overall situation of the $K$ selected indicators.
>
> Existing CD backbones also jointly model the complex interactions among all variates. The main reason of their inferior performance is the overfitting issue, which is caused by their unawareness of the dynamic variation in leading indicators and leading steps. Particularly, LIFT alleviates overfitting by decomposing CD modeling into two steps:
>
> 1) An efficient Lead Estimator roughly ranks potential leading indicators by cross-correlations, and then the indicators are shifted by leading steps;
> 2) A learnable Lead-aware Refiner jointly models the complex interactions between the selected subset of leading indicators and the target variate.
>
> ----
>
> **Q2: When the influence structure is $X^{(1)} \rightarrow X^{(2)} \rightarrow \cdots \rightarrow X^{(C)}$, LIFT will not pick $X^{(C-1)}$ if it has the lowest cross-correlation.**
>
> **A2:** The chain structure $X^{(1)} \rightarrow X^{(2)} \rightarrow \cdots \rightarrow X^{(C)}$ indicates that the leading steps of the $C-1$ indicators *w.r.t.* $X^{(C)}$ are in a descending order, *i.e.*, $\delta _1^{(1)}>\cdots>\delta _{C-2}^{(C)}>\delta _{C-1}^{(C)}$. On the one hand, a leading indicator with a larger leading step offers more advance information. On the other hand, a higher cross-correlation coefficient indicates a more possible lead-lag relationship. Thus, it is better to pick $\{X^{(i)}\} _{i=1}^{C-2}$ which are more helpful and more reliable than $X^{(C-1)}$.
>
> ----
>
> **Q3: Is it possible to discover causal relationships (e.g., Granger causality) among time series via modifications of LIFT?**
>
> **A3:** A possible way to compute a time-dependent causal graph via the modifications of LIST is the following. Initially, we build a graph based on the lead estimation results at time $t$, where each node (a variate $j$) is connected to $K$ nodes (its potential indicators $\mathcal I _t^{(j)}$), *i.e.*, $\{X _{t-\delta _{i, t}^{(j)}}^{(i)} \rightarrow X _t^{(j)} \mid 1 \leq j \leq C, i \in \mathcal I _t^{(j)}\}$. Then, we prune the graph as follows.
>
> 1. Replacing the sum pooling in Eq. (11) by mean pooling, we rewrite our frequency mixing as
>
> $$
> \widetilde{V}^{(j)} = g\left({V}^{(j)}\parallel \frac{1}{|\mathcal K^{(j)}|}\sum _{k\in \mathcal K^{(j)}} {\widetilde U}^{(j)} _k \parallel \frac{1}{|\mathcal K^{(j)}|}\sum _{k\in \mathcal K^{(j)}} {\widetilde \Delta}^{(j)} _k \right),
> $$
>
> where $\mathcal K^{(j)}=\{1,2,\cdots,K\}$ denotes the indices of $\mathcal I _t^{(j)}$.
>
> 2. We train LIFT on the training data and calculate the forecast loss on the validation data.
>
> 3. For each target node $j$ in $[1, C]$, we repeat removing the most useless indicator by the following iteration algorithm until the loss of forecasting $j$ cannot be decreased.
>
>    - For each $k$ in $\mathcal K^{(j)}$, we specially dismiss the $k$-th indicator, re-conduct frequency mixing, and calculate a new forecast loss by using the remaining $|\mathcal K^{(j)}|-1$ indicators. A lower loss indicates that the $k$-th selected one is just a noise.
>
>    - Then, we update $\mathcal K^{(j)}$ as $\mathcal K^{(j)}-\{k^{\prime}\}$ if dismissing the $k^{\prime}$-th indicator leads to the lowest forecast loss.
>
> ----
>
> **Q4: Why there is subscript $U$ and $\Delta$ in equation (9)?**
>
> **A4:** To simplify notations in the following discussion, we leave out the superscript $(j)$ and the subscript $k$. Let $U$ denote the information of one leading indicator, $V$ denote the prediction of the target variate, and $\Delta=U-V$. Eq. (9) generates different filters $r _U$ and $r _{\Delta}$ for $U$ and $\Delta$, respectively.
>
> The reason is that we consider two kinds of relationships:
>
> - $V$ is directly influenced by $U$, and the ground-truth $V$ contains a degree of $U$, *e.g.*, $V _{true} \approx V + r _U\odot U$;
>
> - $V$ is similar to $U$ when they are both influenced by a latent factor, and the ground-truth $V$ is the interpolation between $V$ and $U$, *e.g.*, $V _{true} \approx (1- r _{\Delta})\odot V + r _{\Delta}\odot U=V+ r _{\Delta}\odot\Delta$.
>
> Note that both situations may exist in practice. We thus use different layers to generate $r _U$ and $r _{\Delta}$. Our ablation study in Figure 5b demonstrates the effectiveness of modeling both kinds of influence.

---

> ### Author Response · Authors · 2023-11-21
> **Looking Forward to Your Feedback**
>
> Dear Reviewer MEf1,
>
> Thank you again for your dedication to reviewing our paper! In our early response, we have included more elaboration on our method. We will highly appreciate it if you let us know whether your previous concerns have been adequately addressed.

---

> > ### Comment · Reviewer_MEf1 · 2023-11-22
> > **Response to the authors**
> >
> > I do appreciate the author's response. I will maintain my current score.

---

### Official Review · Reviewer_JmQw · 2023-11-09

**Soundness:** 3 good
**Presentation:** 3 good
**Contribution:** 3 good
**Rating:** 6
**Confidence:** 2

**Summary:**

This paper proposes a new method, called LIFT, which first efficiently estimates leading indicators and their leading steps at each time step, and then judiciously allows the lagged variates to utilize the advance information from leading indicators. This method can be used as a plugin, which is seamlessly collaborated with arbitrary time series forecasting methods. Extensive experiments on six real-world datasets demonstrate the effectiveness of the proposed method with respect to the average forecasting performance.

**Strengths:**

1. This proposed method is somewhat novel, making benefits of the channel dependence. It can efficiently estimate leading indicators and the leading steps, and allow the lagged variates to utilize the advanced information from leading indicators.

2. This method can be regarded as a plugin to collaborate with any other time series forecasting methods.

**Weaknesses:**

1. It remains unclear to me how instance normalization and denormalization could effect the results and performances.

2. In Figure 3, “all layers in the grey background” refer to which part in the figure?

3. Inappropriate expression: “one of the hottest” -> “one the most popular”.

**Questions:**

(See above)

---

> ### Author Response · Authors · 2023-11-15
> **Response to Reviewer JmQw**
>
> Thank you for your time and efforts in reviewing our paper. The followings are our responses to your specific questions.
>
> **Q1: The effect of instance normalization and denormalization.**
>
> **A1:** As a popular preprocessing method in time series forecasting, instance normalization and denormalization is used to avoid disturbance from value ranges. Particularly in our method, instance normalization is an essential prerequisite for computing cross-correlation coefficients as defined in Eq. (1).
> $$
> R^{(j)} _{i,t}(\delta) = \frac{\operatorname{Cov}(\mathcal X^{(i)} _{t-L+1-\delta:t-\delta}, \mathcal X^{(j)} _{t-L+1:t})}{\sigma^{(i)}\sigma^{(j)}}  = \frac{1}{L}\sum _{t^\prime=t-L+1}^{t}{\frac{\mathcal X^{(i)} _{t^\prime-\delta}-\mu^{(i)}}{\sigma^{(i)}}\cdot\frac{\mathcal X^{(j)} _{t^\prime}-\mu^{(j)}}{\sigma^{(j)}}}
> $$
> Intuitively, it is infeasible to directly measure lead-lag relationships by the raw series which have varying value ranges (*e.g.*, one ranging from 1 to 10 and another ranging from 100 to 1000). The advance information from a leading indicator is the shared general up/down trends rather than the specific values. After refining the normalized series of lagged variates by the advance information, it is necessary to restore them to the original value range via denormalization, yielding the final forecasts.
>
> ----
>
> **Q2: In Figure 3, “all layers in the grey background” refer to which?**
>
> **A2:** The grey rectangles refer to normalization, denormalization, Lead Estimator, Fourier transform, and the inverse Fourier transform. We would like to highlight that we avoid repeating Lead Estimation every epoch but precompute it *only once* on training data since our Lead Estimator is non-parametric.
>
> ----
>
> **Q3: Inappropriate expression: “one of the hottest” -> “one of the most popular”.**
>
> **A3:** Thanks for your advice. We have modified the expression in the re-uploaded paper.

---

> > ### Comment · Reviewer_JmQw · 2023-11-22
> > **Response to the authors**
> >
> > Thanks for the reply, which addressed my concerns. I will maintain my positive assessment.

---

### Meta-Review · Area_Chair_jJQt · 2023-12-13

**Metareview:**

This paper is concerned with the problem of multivariate time series forecasting. The authors argue that there exist locally stationary lead-lag relationships between variates, i.e., some lagged variates may follow the leading indicators within a short time period. Exploiting such channel dependence is beneficial since leading indicators offer advance information that can be used to reduce the forecasting difficulty of the lagged variates. They then propose a forecasting algorithm called LIFT that exploits dependencies among the time series by first finding a set of leading time series and their corresponding leading steps for a given time series and then uses a Lead-aware Refiner that adaptively leverages the informative signals of leading indicators in the frequency domain to refine the predictions of lagged variates. Extensive experiments on six real-world datasets demonstrate the effectiveness of the proposed method w.r.t. the average forecasting performance.

In general, the paper considers an important problem and is well written.  The idea is sensible and intuitive, and the empirical results are very encouraging.

**Justification For Why Not Higher Score:**

The novelty of the idea is good, but not impressive.

**Justification For Why Not Lower Score:**

In general, the paper considers an important problem and is well written.  The idea is sensible and intuitive, and the empirical results are very encouraging.

---

### Decision · Program_Chairs · 2024-01-16

Accept (poster)